# Linking In-context Learning in Transformers to Human Episodic Memory

**Li Ji-An** *
Neurosciences Graduate Program
University of California, San Diego
jil095@ucsd.edu

**Corey Y Zhou** *
Department of Cognitive Science
University of California, San Diego
yiz329@ucsd.edu

**Marcus K. Benna** †
Department of Neurobiology
University of California, San Diego
mbenna@ucsd.edu

**Marcelo G. Mattar** †
Department of Psychology
New York University
marcelo.mattar@nyu.edu

## Abstract

Understanding connections between artificial and biological intelligent systems can reveal fundamental principles of general intelligence. While many artificial intelligence models have a neuroscience counterpart, such connections are largely missing in Transformer models and the self-attention mechanism. Here, we examine the relationship between interacting attention heads and human episodic memory. We focus on induction heads, which contribute to in-context learning in Transformer-based large language models (LLMs). We demonstrate that induction heads are behaviorally, functionally, and mechanistically similar to the contextual maintenance and retrieval (CMR) model of human episodic memory. Our analyses of LLMs pre-trained on extensive text data show that CMR-like heads often emerge in the intermediate and late layers, qualitatively mirroring human memory biases. The ablation of CMR-like heads suggests their causal role in in-context learning. Our findings uncover a parallel between the computational mechanisms of LLMs and human memory, offering valuable insights into both research fields.

## 1 Introduction

Neural networks often bear striking similarities to biological intelligence. For instance, convolutional networks trained on computer vision tasks can predict neuronal activities in the visual cortex [1–4]. Recurrent neural networks trained on spatial navigation develop neural representations similar to the entorhinal cortex and hippocampus [5, 6], and those trained on reward-based tasks recapitulate biological decision-making behaviors [7]. Feedforward networks trained on category learning exhibit human-like attentional bias [8]. Identifying commonalities between artificial and biological intelligence offers unique insights into both model properties and the brain's cognitive functions.

In contrast to this long tradition of drawing parallels between AI models and biology, exploration of the biological relevance of the Transformer architecture — originally proposed for natural language translation [9] — remains limited, with many researchers in neuroscience, cognitive science, and deep learning viewing it as being fundamentally different from the brain. So far, Transformers have been linked to generalized Hopfield networks [10], neural activities in the language cortex [11–13],

---

*Equal contribution.
†Co-senior author.

38th Conference on Neural Information Processing Systems (NeurIPS 2024).

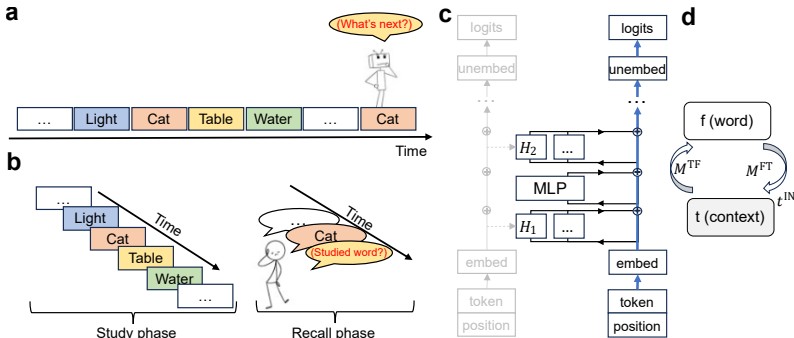

Figure 1: **Tasks and model architectures. (a)** Next-token prediction task. The ICL of pre-trained LLMs is evaluated on a sequence of repeated random tokens ("...[A][B][C][D]...[A][B][C][D]..."; e.g., [A]=light, [B]=cat, [C]=table, [D]=water) by predicting the next token (e.g., "...[A][B][C][D] ...[B]"→ ?). **(b)** Human memory recall task. During the study phase, the subject is sequentially presented with a list of words to memorize. During the recall phase, the subject is required to recall the studied words in any order. **(c)** Transformer architecture, centering on the residual stream. The blue path is the residual stream of the current token, and the grey path represents the residual stream of a past token. $H_1$ and $H_2$ are attention heads. MLP is the multilayer perceptron. **(d)** Contextual maintenance and retrieval model. The word vector $\mathbf{f}$ is retrieved from the context vector $\mathbf{t}$ via $\mathbf{M}^{\mathrm{TF}}$ and the context vector is updated by the word vector via $\mathbf{M}^{\mathrm{FT}}$ (see main text for details).

and hippocampo-cortical circuit representations [14]. However, it remains unclear whether and how the *emergent* behavior and mechanisms of *interacting* attention heads relate to biological cognition.

This study bridges this gap by examining the parallels between attention heads in Transformer models and episodic memory in biological cognition. We focus on "induction heads", a particular type of attention head in Transformer models and a crucial component of *in-context learning* (ICL) observed in LLMs [15]. ICL enables LLMs to perform new tasks on the fly during test time, relying solely on the context provided in the input prompt, without the need for additional fine-tuning or task-specific training [16, 17]. We show that induction heads share several parallel properties with the contextual maintenance and retrieval (CMR) model, an influential model of human episodic memory during free recall. Understanding the mechanisms of ICL is important for developing better models capable of performing unseen tasks, as well as for AI safety research, as the models could be instructed to perform malicious activities after being deployed in real-world scenarios.

In sections below, we introduce the task in Section 2, Transformer and induction heads in Sections 3.1, 3.2, and the CMR model in Section 4.1. We show that induction heads and CMR are *mechanistically* similar in Sections 3.3 and 4.2 and *behaviorally* similar in Section 5.1. We further characterize the emergence of CMR-like behavior in Section 5.2 and its possible causal role in Section 5.3. Overall, our study provides evidence for a novel bridge between Transformer models and episodic memory.

## 2 Next-token prediction and memory recall

Transformer models in language modeling are often trained to predict the next token [16]. ICL thus helps next-token prediction using information provided solely in the input prompt context. One way to evaluate a model's ICL is to run it on a sequence of repeated random tokens [15] (Fig. 1a). For example, consider the prompt "[A][B][C][D][A][B][C][D]". Assuming that no structure between these tokens has been learned, the first occurrence of each token cannot be predicted — e.g., the first [C] cannot be predicted to follow the first [B]. At the second [B], however, a model with ICL should predict [C] to follow by retrieving the temporal association in the first part of the context.

Much like ICL in a Transformer model, human cognition is also known to perform associative retrieval when recalling episodic memories. Episodic retrieval is commonly studied using the free recall paradigm [18, 19] (Fig. 1b). In free recall, participants first study a list of words sequentially, and are then asked to freely recall the studied words in any order [20]. Despite no requirements on recall order, humans often exhibit patterns of recall that reflect the temporal structure of the preceding study list. In particular, the retrieval of one word triggers the subsequent recall of other words studied close in time (temporal contiguity). Additionally, words studied *after* the previously recalled word are

retrieved with higher probability than words studied *before* the previously recalled word, leading to a tendency of recalling words in the same temporal ordering of the study phase (forward asymmetry). These effects are typically quantified through the conditional response probability (CRP): given the most recently recalled stimulus with a serial position $i$ during study, the CRP is the probability that the subsequently recalled stimulus comes from the serial position $i$+lag (see e.g., Fig. 4a).

## 3 Transformer models and induction heads

### 3.1 Residual stream and interacting heads

The standard view of Transformers emphasizes the stacking of Transformer blocks. An alternative, mathematically equivalent view emphasizes the *residual stream* [21, 22]. Each token at position $i$ in the input has its own residual stream $z_i$ (with a dimension of $d_{\text{model}}$) serving as a shared communication channel between model components at different layers (Fig. 1c; the residual stream is shown as a blue path), such as self-attention and multi-layered perceptrons (MLP). The initial residual stream $z_i^{(0)}$ contains *token embeddings* $TE$ (vectors that represent tokens in the semantic space) and *position embeddings* $PE$ (vectors that encode positions of each input token). Each model component reads from the residual stream, performs a computation, and *additively* writes into the residual stream. Specifically, attention heads at layer $l$ read from all past $z_j$ (with $j \leq i$) and write into the current $z_i$ as $z_i^{(l)'} \leftarrow z_i^{(l-1)} + \sum_{\text{heads } h} H^{(h)}(z_i^{(l-1)}; \{z_j^{(l-1)}\}_{j \leq i})$, while MLP layers read from only the current $z_i$ and write into $z_i$ as $z_i^{(l)} \leftarrow z_i^{(l)'} + \text{MLP}(z_i^{(l-1)'})$. Readers unfamiliar with attention heads (e.g., attention scores and patterns) are referred to Appendix B. Other components like layer normalization are omitted for simplicity. At the final layer, the residual stream is passed through the unembedding layer to generate the logits (input to softmax) that predict the next token.

Components in different layers can interact with each other through the residual stream [21]. As an important example, a first-layer head $H_1$ may write its output into the residual stream, which is later read by a second-layer head $H_2$ that writes its output to the residual stream for later layers to use.

### 3.2 Induction heads and their attention patterns

Previous mechanistic interpretability studies identified a type of attention heads critical for ICL, known as *induction heads* [21, 15, 23, 24]. Induction heads are defined by their *match-then-copy* behavior [15, 24]. They look back (*prefix matching*) over previous occurrences of the current input token (e.g., [B]), determine the subsequent token (e.g., [C] if the past context included the pair [B][C]), and increase the probability of the latter – that is, after finding a "match", it makes a "copy" as the predicted next token (. . . [B][C] . . . [B] → [C]). To formalize this match-then-copy pattern, we use the induction-head matching score (between 0 and 1) to measure the prefix-matching behavior. We then use the copying score (between -1 and 1) to measure the copying behavior (see Appendix E). An induction head should have a large induction-head matching score and a positive copying score.

We first examined the induction behaviors of attention heads in the pre-trained GPT2-small model [16] using the TransformerLens library [25]. To elicit induction behaviors, we constructed a prompt consisting of two repeats of a random-token sequence (see Section 2 and Appendix D). We recorded the attention scores of each head $\tilde{\mathbf{A}}$ (before softmax) and the attention patterns $\mathbf{A}$ (after softmax) for each pair of previous and current token positions (for definitions see Appendix B). Several heads in GPT2-small had a high induction-head matching score (Fig. 2a), such as L5H1 (layer number 5, head number 1). In the first sequence repeat, this attention head attends mostly to the beginning-of-sequence token. In the second repeat, this head shows a clear "induction stripe" (Fig. 2b) where it mostly attends to the token that follows the current token in the first repeat.

We calculated attention scores as a function of relative position lags to further characterize the behavior of induction heads. This analysis is reminiscent of the CRP analysis on human recall data, as in Section 4.1. We found that induction heads' attention to earlier tokens follows a similar pattern as seen in human episodic recall (Fig. 2c, Fig. 5a-c), including temporal contiguity (e.g., the average attention score for |lag| $\leq 2$ is larger than for |lag| $\geq 4$) and forward asymmetry (e.g., the average attention score for lag$> 0$ is larger than for lag$< 0$).

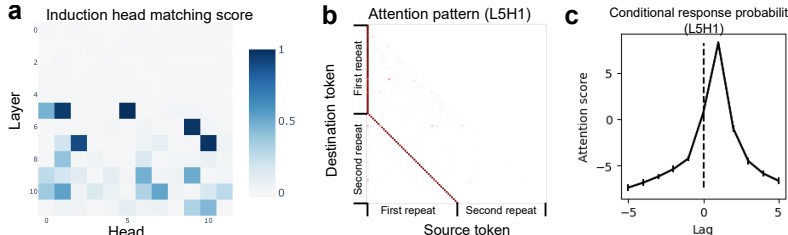

Figure 2: **Induction heads in the GPT2-small model. (a)** Several heads in GPT2 have a relatively large induction-head matching score. **(b)** The attention pattern of the L5H1 head, which has the largest induction-head matching score. The diagonal line ("induction stripe") shows the attention from the destination token in the second repeat to the source token in the first repeat. **(c)** The attention scores of the L5H1 head averaged over all tokens in the designed prompt as a function of the relative position lag (similar to CRP). Error bars show the SEM across tokens.

### 3.3 K-composition and Q-composition induction heads

The matching score and copying score describe the behavior of individual attention heads. However, they do not provide a mechanistic understanding of *how* the induction head works internally. To gain insights into the internal mechanisms of induction heads, we focus here on smaller transformer models, acknowledging that individual attention heads of larger LLMs likely exhibit more sophisticated behavior. Prior work has discovered two kinds of induction mechanisms in two-layer attention-only Transformer models: K-composition and Q-composition (Fig. 3a-b, Tab. S1) [21, 24], characterizing how information from the first-layer head is composed to inform attention of the second-layer head. We provide an overview of both below. Our main focus in this paper is the comparison between Q-composition and CMR, but K-composition is provided as a point of comparison for readers familiar with mechanistic interpretability.

In K-composition (Fig. 3a), the first-layer "previous token" head uses the current token's position embedding, $PE_i$, as the query, and a past token's position embedding $PE_j$, as the key. When the match condition $PE_j = PE_{i-1}$ is satisfied (meaning $j$ is the previous position of $i$), the head writes the previous token's token embedding, $TE_j$, as the value into the residual stream $z_i$. The second-layer induction head uses the current token's $TE_k$ as the query, and the previous token head's output $TE_j$ at residual stream $z_i$ as the key ("K-composition"). When the match condition $TE_j = TE_k$ is satisfied, the head writes $TE_i$ (at residual stream $z_i$) as the value into the residual stream $z_k$, effectively increasing the logit for the token that occurred at position $i$.

In Q-composition (Fig. 3b), the first-layer "duplicate token" head uses the current token's $TE_k$ as the query, and a past token's $TE_j$ as the key. When the match condition $TE_j = TE_k$ is satisfied (meaning token $k$ is a duplicate of token $j$), the head writes the token's $PE_j$ as the value into the residual stream $z_k$. The second-layer induction head uses the duplicate token head's output $PE_j$ at residual stream $z_k$ as the query ("Q-composition") and a past token's $PE_i$ as the key. When the match condition $PE_j = PE_{i-1}$ is satisfied, the head writes $TE_i$ (at residual stream $z_i$) as the value into the residual stream $z_k$, increasing the logit for the token that occurred at position $i$.

In the following sections, we will reveal a novel connection between ICL and human episodic memory. We first introduce the CMR model of episodic memory, and then formally re-write it as a Q-composition induction head performing prefix matching, allowing us to link induction heads' attention biases to those known in human episodic memory.

## 4 Contextual maintenance and retrieval model (CMR)

### 4.1 CMR in its original form

CMR, an influential model of human episodic memory, provides a general framework to model memory recall as associative retrieval. It leverages a distributed representation called *temporal context*, which acts as a dynamic cue for recalling subsequent information based on recently seen words [26]. CMR explains the asymmetric contiguity bias in human free recall (see Fig. 4 and Fig. S1) and has been extended to more complex memory phenomena such as semantic [27] and emotional

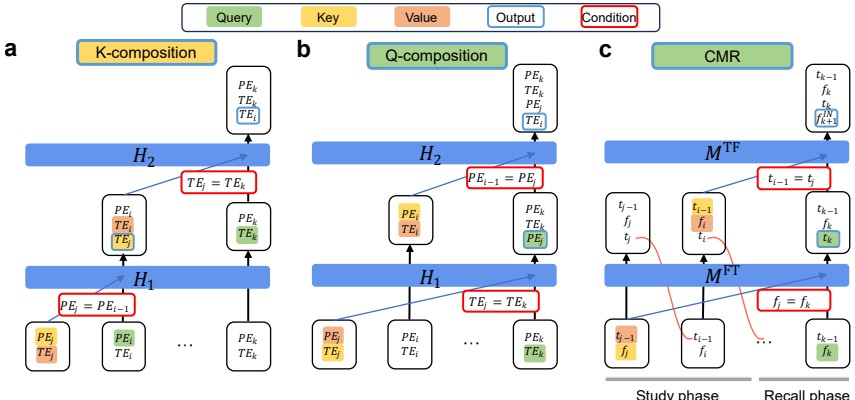

Figure 3: **Comparison of composition mechanisms of induction heads and CMR.** All panels correspond to the optimal Q-K match condition ($j = i - 1$). See the main text and Tab. S1 for details. **(a)** K-composition induction head. The first-layer head's output serves as the *Key* of the second-layer head. **(b)** Q-composition induction head. The first-layer head's output serves as the *Query* of the second-layer head. **(c)** CMR is similar to a Q-composition induction head, except that the context vector $t_{j-1}$ is first updated by $\mathbf{M}^{\mathrm{FT}}$ into $t_j$ at position $j$, then directly used at position $j + 1$ (equal to $i$ for the optimal match condition; shown by red lines).

[28] effects. We provide a pedagogical introduction to CMR in Appendix C for readers without a background in cognitive science or episodic memory. Below, we briefly list the essentials of CMR.

In CMR (Fig. 1d), each word token is represented by an embedding vector $\mathbf{f}$ (e.g., one-hot; $\mathbf{f}_i$ for the $i$-th word in a sequence). The core dynamic that drives both sequential encoding and retrieval is

$$\mathbf{t}_i = \rho \mathbf{t}_{i-1} + \beta \mathbf{t}_i^{\mathrm{IN}}, \tag{1}$$

where $\mathbf{t}_i$ is the temporal context at time step $i$, and $\mathbf{t}_i^{\mathrm{IN}}$ is an input context associated with $\mathbf{f}_i$. $\beta$ controls the degree of *temporal drift* between time steps ($\beta_{\mathrm{enc}}$ for encoding/study phase and $\beta_{\mathrm{rec}}$ for decoding/retrieval phase) and $\rho$ is picked to ensure $\mathbf{t}_i$ has unit norm. Specifically, during the encoding phase, $\mathbf{t}_i^{\mathrm{IN}}$ represents a *pre*-experimental context associated with the $i$-th word as $\mathbf{t}_i^{\mathrm{IN}} = \mathbf{M}_{\mathrm{pre}}^{\mathrm{FT}} \mathbf{f}_i$, where $\mathbf{M}_{\mathrm{pre}}^{\mathrm{FT}}$ is a pre-fixed matrix. At each time step, a word-to-context (mapping $\mathbf{f}$ to $\mathbf{t}$) memory matrix $\mathbf{M}_{\mathrm{exp}}^{\mathrm{FT}}$ learns the association between $\mathbf{f}_i$ and $\mathbf{t}_{i-1}$ (i.e., $\mathbf{M}_{\mathrm{exp}}^{\mathrm{FT}}$ is updated by $\mathbf{t}_{i-1} \mathbf{f}_i^T$). During the decoding (retrieval) phase, $\mathbf{t}_i^{\mathrm{IN}}$ is a mixture of pre-experimental ($\mathbf{t}_{\mathrm{pre}}^{\mathrm{IN}} = \mathbf{M}_{\mathrm{pre}}^{\mathrm{FT}} \mathbf{f}_i$) and experimental contexts ($\mathbf{t}_{\mathrm{exp}}^{\mathrm{IN}} = \mathbf{M}_{\mathrm{exp}}^{\mathrm{FT}} \mathbf{f}_i$). The proportion of these two contexts is controlled by an additional parameter $\gamma_{\mathrm{FT}} \in [0, 1]$ as $\mathbf{t}_i^{\mathrm{IN}} = ((1 - \gamma_{\mathrm{FT}}) \mathbf{M}_{\mathrm{pre}}^{\mathrm{FT}} + \gamma_{\mathrm{FT}} \mathbf{M}_{\mathrm{exp}}^{\mathrm{FT}}) \mathbf{f}_i$. The asymmetric contiguity bias arises from this slow evolution of temporal context: when $0 < \beta < 1$, $\mathbf{t}_i$ passes through multiple time steps, causing nearby tokens to be associated with temporally adjacent contexts that are similar to each other (temporal contiguity), i.e., $\langle \mathbf{t}_i, \mathbf{t}_j \rangle$ is large if $|i - j|$ is small. Additionally, $\mathbf{t}_{\mathrm{pre}}^{\mathrm{IN}} = \mathbf{M}_{\mathrm{exp}}^{\mathrm{FT}} \mathbf{f}_i$ only enters the temporal context after time $i$. Thus $\mathbf{t}_{i,\mathrm{exp}}^{\mathrm{IN}}$ is associated with $\mathbf{f}_j$ *only* for $j > i$ (asymmetry).

CMR also learns a second context-to-word (mapping $\mathbf{t}$ back to $\mathbf{f}$) memory matrix $\mathbf{M}^{\mathrm{TF}}$ (updated by each $\mathbf{f}_i \mathbf{t}_{i-1}^T$). When an output is needed, CMR retrieves a mixed word embedding $\mathbf{f}_i^{\mathrm{IN}} = \mathbf{M}^{\mathrm{TF}} \mathbf{t}_i$. If $\mathbf{f}_j$ are one-hot encoded, we can simply treat $\mathbf{f}_i^{\mathrm{IN}}$ as a (unnormalized) probability distribution over the input tokens. Or, CMR can compute the inner product $\langle \mathbf{f}_j, \mathbf{f}_i^{\mathrm{IN}} \rangle$ for each cached word $\mathbf{f}_j$ as input to softmax (with an inverse temperature $\tau^{-1}$) to recall a word.

Intuitively, the temporal context resembles a moving spotlight with a fuzzy edge: it carries recency-weighted historical information that may be relevant to the present, where the degree of information degradation is controlled by $\rho$. Larger $\beta$'s correspond to "sharper" CRPs with stronger forward asymmetry and stronger temporal clustering that are core features of human episodic memory. As a concrete example, consider $n$ unique one-hot encoded words $\{\mathbf{f}_i\}$. If $\mathbf{M}_{\mathrm{pre}}^{\mathrm{FT}} = \sum_{i=1}^{n} \mathbf{f}_i \mathbf{f}_i^T$ (i.e., the pre-experimental context associated with each word embedding is the word embedding itself) and $\gamma_{\mathrm{FT}} = 0$, Eq. 1 at decoding is reduced to $\mathbf{t}_i = \rho \mathbf{t}_{i-1} + \beta \mathbf{f}_i = \sum_{j=1}^{i} \beta \rho^{i-j} \mathbf{f}_j$, which is a linear combination of past word embeddings.

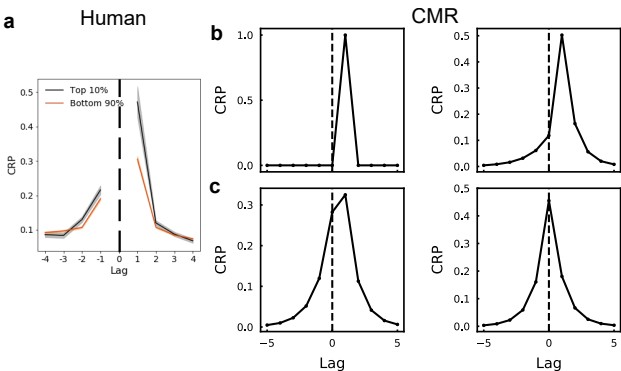

Figure 4: **The conditional response probability (CRP) as a function of position lags in a human experiment and different parametrization of CMR. (a)** CRP of participants (N=171) in the PEERS dataset, reproduced from [29]. "Top 10%" refers to participants whose performance was in the top 10th percentile of the population when recall started from the beginning of the list. They have a sharper CRP with a larger forward asymmetry than other subjects. **(b)** Left, CMR with "sequential chaining" behavior ($\beta_{\text{enc}} = \beta_{\text{rec}} = 1, \gamma_{\text{FT}} = 0$). The recall has exactly the same order as the study phase without skipping over any word. Right, CMR with moderate updating at both encoding and retrieval, resulting in human-like free recall behavior ($\beta_{\text{enc}} = \beta_{\text{rec}} = 0.7, \gamma_{\text{FT}} = 0$). Recall is more likely than not to have the same order as during study and sometimes skips words. **(c)** Same as (b Right) except with $\gamma_{\text{FT}} = 0.5$ (Left) and $\gamma_{\text{FT}} = 1$ (Right). For more examples, see Fig. S1.

## 4.2 CMR as an induction head

We now proceed to map CMR to a Q-composition-like head (see Fig. 3c and Tab. S1 for details).

To begin with, we note that the word $\mathbf{f}_i$ seen at position $i$ is the same as $TE_i$, and the context vector $\mathbf{t}_{i-1}$ (before update) at position $i$ is functionally similar to $PE_i$. It follows that the set $\{\mathbf{f}_i, \mathbf{t}_{i-1}\}$ is functionally similar to the residual stream $z_i$, updated by the head outputs.

**CMR experimental word-context retrieval as first-layer self-attention**. The temporal context is updated by $\mathbf{t}_i^{\text{IN}} = ((1 - \gamma_{\text{FT}})\mathbf{M}_{\text{pre}}^{\text{FT}} + \gamma_{\text{FT}}\mathbf{M}_{\text{exp},i}^{\text{FT}})\mathbf{f}_i$ at decoding. The memory matrix $\mathbf{M}_{\text{exp},i}^{\text{FT}}$ acts as a first-layer duplicate token head, where the current word $\mathbf{f}_i$ is the query, the past embeddings $\{\mathbf{f}_j\}$ make up the keys, and the temporal contexts $\mathbf{t}_{j-1}$ associated with each $\mathbf{f}_j$ are values. This head effectively outputs "What's the position (context vector) at which I encountered the same token $\mathbf{f}_i$?"

**CMR pre-experimental word-context retrieval as MLP**. The pre-experimental context $\mathbf{t}_i^{\text{IN}} = \mathbf{M}_{\text{pre}}^{\text{FT}}\mathbf{f}_i$ (retrieved contextual information not present in the experiment) is the output of a linear fully-connected layer (functionally similar to MLP; not drawn).

**CMR evolution as residual stream updating**. The context vector is updated by $\mathbf{t}_k = \rho\mathbf{t}_{k-1} + \beta\mathbf{t}_k^{\text{IN}}$. Equivalently, the head $\mathbf{M}^{\text{FT}}$ updates the information from $\{\mathbf{f}_k, \mathbf{t}_{k-1}\}$ to $\{\mathbf{f}_k, \mathbf{t}_{k-1}, \mathbf{t}_k\}$. At the position $k$ during recall, the updated context $\mathbf{t}_k$ contains $\mathbf{t}_k^{\text{IN}}$ ($\approx \mathbf{t}_j$) (Fig. 3c).

**CMR context-word retrieval as second-layer self-attention**. The retrieved embedding is $\mathbf{f}_k^{\text{IN}} = \mathbf{M}^{\text{TF}}\mathbf{t}_k$, where $\mathbf{M}^{\text{TF}}$ acts as a second-layer induction head, where the temporal context $\mathbf{t}_k$ is the query, the past contexts $\{\mathbf{t}_{i-1}\}$ make up the keys, and the embeddings $\mathbf{f}_i$ associated with each $\mathbf{t}_{i-1}$ are values. This effectively implements Q-composition [21], because $\mathbf{t}_k$, as the *Query*, is affected by the output of the first-layer $\mathbf{M}_{\text{exp}}^{\text{FT}}$ head.

**CMR word recall as unembedding**. The final retrieved word probability is determined by the inner product between the retrieved memory $\mathbf{f}_k^{\text{IN}}$ and each studied word $\mathbf{f}_j$, similar to the unembedding layer generating the output logits from the residual stream.

**CMR learning as a causal linear attention head**. Both associative matrices of CMR are learned in a manner consistent with the formation of causal linear attention heads. Specifically, the word-to-context matrix is updated by $\mathbf{M}_{\text{exp},i}^{\text{FT}} = \mathbf{M}_{\text{exp},i-1}^{\text{FT}} + \mathbf{t}_{i-1}\mathbf{f}_i^T$ (with $\mathbf{M}_{\text{exp},0}^{\text{FT}} = \mathbf{0}$), associating $\mathbf{f}_i$ (key) and $\mathbf{t}_{i-1}$ (value). It is equivalent to a causal linear attention head, because $\mathbf{M}_{\text{exp}}^{\text{FT}}\mathbf{f}_k = (\sum_{i<k}\mathbf{t}_{i-1}\mathbf{f}_i^T)\mathbf{f}_{\mathbf{k}} = \sum_{i<k}\mathbf{t}_{i-1}(\mathbf{f}_i^T\mathbf{f}_{\mathbf{k}}) = \sum_{i<k}\text{sim}(\mathbf{f}_i, \mathbf{f}_{\mathbf{k}})\mathbf{t}_{i-1}$. Similarly, the context-to-word

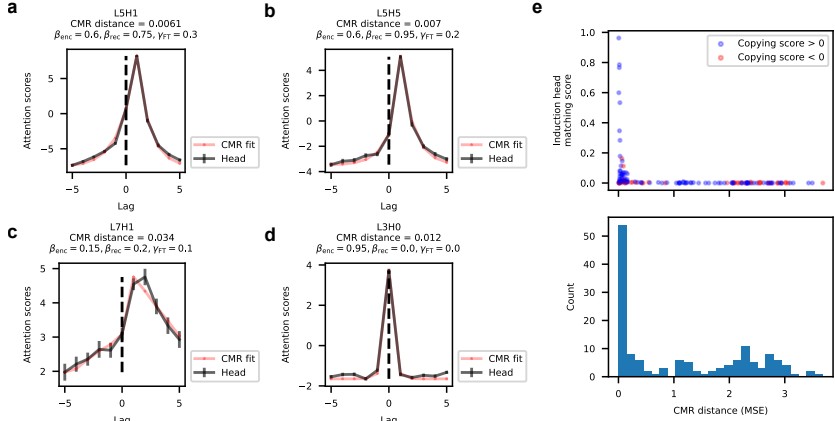

Figure 5: **CMR distance provides meaningful descriptions for attention heads in GPT2. (a-c)** Average attention scores and the CMR-fitted attention scores of example induction heads (with a non-zero induction-head matching score and positive copying score). **(d)** Average attention scores and the CMR-fitted attention scores of a duplicate token head [30] that is traditionally not considered an induction head but can be well-captured by the CMR. **(e)** (Top) CMR distance (measured by MSE) and the induction-head matching score for each head. (Bottom) Histogram of the CMR distance.

matrix, updated by $\mathbf{M}_i^{\mathrm{TF}} = \mathbf{M}_{i-1}^{\mathrm{TF}} + \mathbf{f}_i \mathbf{t}_{i-1}^T$ (with $\mathbf{M}_0^{\mathrm{TF}} = \mathbf{0}$), is equivalent to a causal linear attention head that associates $\mathbf{t}_{i-1}$ (key) with $\mathbf{f}_i$ (value).

To summarize, the CMR architecture resembles a two-layer transformer with a Q-composition linear induction head. It's worth noting that although we cast $\mathbf{t}_i$ as the position embedding, unlike position embeddings that permit parallel processing in Transformer models, $\mathbf{t}_i$ is recurrently updated in CMR (Eq. 1). It is possible that Transformer models might acquire induction heads with a similar circuit mechanism, where $\mathbf{t}_i$ corresponds to autoregressively updated context information in the residual stream that serves as the input for downstream attention heads.

## 5 Experiments

### 5.1 Quantifying the similarity between an induction head and CMR

We have shown that induction heads in pre-trained LLMs exhibit CMR-like attention biases (Fig. 2c, Fig. 5a-c; also see Fig. S2 for heads unlike CMR) and established the mechanistic similarity between induction heads and CMR (Fig. 3). To further quantify their behavioral similarity, we propose the metric *CMR distance*, defined as the mean squared error (MSE) between the head's average attention scores and its CMR-fitted scores (see Appendix E, Fig. 5a-d). In essence, we optimized the parameters $(\beta_{\mathrm{enc}}, \beta_{\mathrm{rec}}, \gamma_{\mathrm{FT}}, \tau^{-1})$ for each head to obtain a set of CMR-fitted scores that minimizes MSE.

At the population level, heads with a large induction-head matching score and a positive copying score also have a smaller CMR distance (Fig. 5e), suggesting that the CMR distance captures meaningful behavior of these heads. Notably, certain heads that are not typically considered induction heads (e.g., peaking at lag=0) can be well captured by CMR (Fig. 5d).

Consistent with prior findings that induction heads were primarily observed in the intermediate layers of LLMs [31], we found that the majority of heads in the intermediate-to-late layers of GPT2-small have lower CMR distances (Fig. 6a, Fig. S3a; for the choice of threshold see Appendix E.5). We also observed similar phenomena in a different set of LLMs called Pythia (Fig. 6b), a family of models with shared architecture but different sizes, as well as three well-known models (Qwen-7B [32], Mistral-7B [33], Llama3-8B [34], Fig. 6c). We summarized these results in Fig. S3b.

Additionally, to contextualize these CMR distances, we included the Gaussian distance using Gaussian functions as a baseline (with the same number of parameters as CMR), since its bell shape captures the basic aspects of temporal contiguity and forward/backward asymmetry. We found that, across 12 different models (GPT2, Pythia models, Qwen-7B, Mistral-7B, Llama3-8B), CMR provides significantly better descriptions (lower distances) than the Gaussian function for the top induction heads (average CMR distance: 0.11 (top 20), 0.05 (top 50), 0.12 (top 100), 0.12 (top 200); average

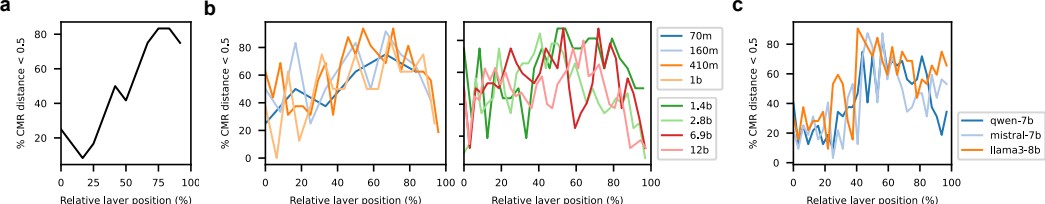

Figure 6: **CMR distances vary with relative layer positions in LLMs. (a-b)** Percentage of heads with a CMR distance less than 0.5 in different layers. Also see Fig. S3c-d for the threshold of 0.1. **(a)** GPT2-small. **(b)** Pythia models across different model sizes (label indicates the number of model parameters). CMR distances are computed based on the last model checkpoint. **(c)** Qwen-7B, Mistral-7B, and Llama-8B models. Heads with lower CMR distances often emerge in the intermediate-to-late layers.

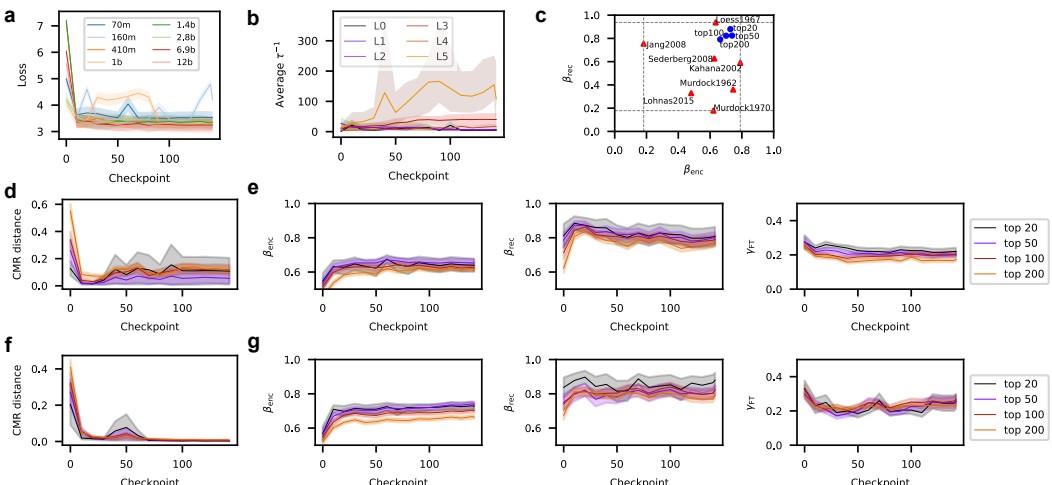

Figure 7: **Strong asymmetric contiguity bias arises as model performance improves. (a)** Model loss on the designed prompt as a function of training time. Loss is recorded every 10 training checkpoints. **(b)** Average fitted inverse temperature increases in the intermediate layers of Pythia-70m as training progresses. Values are averaged across heads with CMR distance lower than 0.5 in each layer. **(c)** Comparison of fitted $\beta_{\mathrm{enc}}$ and $\beta_{\mathrm{rec}}$ in Pythia's top CMR-like heads and in existing human studies. **(d)** CMR distance of top induction heads in Pythia models as a function of training time. Heads are selected based on the highest induction-head matching scores across all Pythia models (e.g., "top 20" corresponds to twenty heads with the highest induction-head matching scores). **(e)** Fitted CMR temporal drift parameters $\beta_{\mathrm{enc}}$(left), $\beta_{\mathrm{rec}}$ (middle), $\gamma_{\mathrm{FT}}$ (right) as a function of training time in attention heads with the highest induction-head matching scores. **(f-g)** Same as c-d but for top CMR-like heads (e.g., "top 20" corresponds to those with the lowest CMR distances), demonstrating differences between top induction heads and top CMR-like heads. Shaded regions indicate standard error, except **(b)** which indicates the range (the scale factor $\tau^{-1}$ is non-negative).

Gaussian distance: 1.0 (top 20), 0.98 (top 50), 0.98 (top 100), 0.97 (top 200); all $p < 0.0001$), again suggesting that CMR is well-poised to explain the properties of observed CRPs.

## 5.2 CMR-like heads develop human-like temporal clustering over training

The Pythia models' different pretraining checkpoints allowed us to measure the emergence of CMR-like behavior. As the model's loss on the designed prompt decreases through training (Fig. 7a), the degree of temporal clustering increases, especially in layers where induction heads usually emerge. For instance, intermediate layers of Pythia-70m (e.g., L3, L4) show the strongest temporal clustering that persists over training (Fig. S4a-b). This, combined with an increasing inverse temperature (Fig. 7b), suggests that attention patterns become more deterministic over training, while shaped to mirror human-like asymmetric contiguity biases. In fact, human subjects with better free recall performance tend to exhibit stronger temporal clustering and a higher inverse temperature [29].

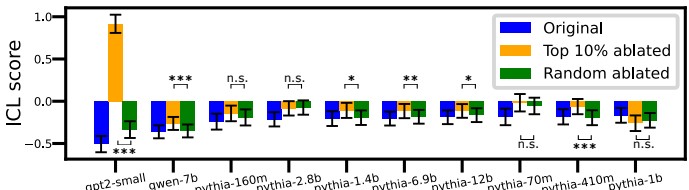

Figure 8: **CMR-like heads are causally relevant for ICL.** ICL scores are evaluated for intact models (Original), models with the top 10% CMR-like heads ablated (Top 10% ablated), and models with randomly selected heads ablated (Random ablated). Lower scores indicate better ICL abilities, with error bars showing SEM across sequences. $***$: $p < 0.001$, $**$: $p < 0.01$, $*$: $p < 0.05$, n.s.: $p \geq 0.1$.

For individual heads, those with higher induction-head matching scores (Fig. 7d) (or similarly with smaller CMR distances, see Fig. 7f) consistently exhibit greater temporal clustering (Fig. 7e, f respectively), as the fitted $\beta$'s (both $\beta_{\text{enc}}$ and $\beta_{\text{rec}}$) were large. We also observed similar values of fitted $\beta$s in Qwen-7B, Mistral-7B, and Llama3-8B (Fig. S6b). These fitted $\beta$'s of these attention heads fall into a similar range as human recall data (Fig. 7c). We interpret this in light of a normative view of the human memory system: in humans, the asymmetric contiguity bias with a $\beta < 1$ is not merely phenomenological; under the CMR framework, it gives rise to an optimal policy to maximize memory recall when encoding and retrieval are noisy [29]. In effect, a large $\beta$ (but less than 1) in Eq. 1 provides meaningful associations beyond adjacent words to facilitate recall, such that even if the immediately following token is poorly encoded or the agent fails to decode it, information from close-by tokens encapsulated in the temporal context still allows the agent to continue decoding.

In addition, we observed an increase in $\beta_{\text{enc}}$ and $\beta_{\text{rec}}$ (Fig. 7e, g) during the first 10 training checkpoints, when the model loss significantly drops. The training process leads to higher values of $\beta_{\text{rec}}$. Specifically, $\beta_{\text{rec}}$ values are higher than $\beta_{\text{enc}}$, highlighting the importance of temporal clustering during decoding for model performance. These results suggest that attention to temporally adjacent tokens with an asymmetric contiguity bias may support ICL in LLMs.

### 5.3 CMR-like heads are causally relevant for ICL capability

While these CMR-like heads were identified using repeated random tokens, we ask whether they were also causally necessary for ICL in more naturalistic tasks. We thus performed an ablation study using natural language texts. Specifically, we ablated either the top 10% CMR-like heads (i.e., top 10% heads with the smallest CMR distances) or the same number of randomly selected heads in each model. We then computed the resultant ICL score on the sampled texts (2000 sequences, each with at least 512 tokens) from the processed version of Google's C4 dataset [35]. The ICL score is defined as the loss of the 500th token in the context minus the loss of the 50th token in the context, averaged over dataset examples [15]. Intuitively, a model with better in-context learning ability has a lower ICL score, as the 500th token is further into the context established from the beginning. We tested models with various model architectures and complexity, including GPT2, Pythia models, and Qwen-7B (Fig. 8). Most models showed a higher ICL score (worse ICL ability) if the top 10% CMR-like heads were ablated, compared to if the same number of randomly selected heads were ablated. This effect was particularly significant if the original model had a low ICL score (e.g., GPT2, Qwen-7B).

Our finding therefore suggests that CMR-like heads are not merely an epiphenomenon, but essential underlying LLM's ICL ability, and that the CMR distance is a meaningful metric to characterize individual heads in LLMs. Nonetheless, our result needs to be interpreted cautiously: First, a Hydra effect has been noted where ablation of heads causes other heads to compensate [36]. Second, our analysis cannot confirm a direct causal role of the CMR-like behavior of these CMR-like heads in ICL, since it is possible that characteristics other than the episodic-memory features in these CMR-like heads might causally contribute to ICL. Finally, the ICL scores for the original Pythia models are closer to 0 (worse ICL performance) than other models', suggesting either weaker ICL ability of the Pythia series, or larger distributional differences between their training data and our evaluation texts.

## 6 Discussion

This study bridges LLMs and human episodic memory by comparing Transformer models' induction heads and CMR. We revealed mechanistic similarities between CMR and Q-composition induction

heads and identified CMR-like attention biases (i.e., asymmetric contiguity) in pre-trained LLMs. Notably, CMR-like heads emerge in LLMs' intermediate-to-late layers, evolve towards a state akin to human memory biases, and may play a causal role in ICL. These findings offer significant connections between the current generation of AI algorithms and a century of human memory research.

From a neuroscience and cognitive science perspective, CMR's link to induction heads might reveal normative principles of memory and hippocampal processing, echoing the role of the hippocampus in pattern prediction and completion [37–39]. While CMR is a behavioral model, it also explains neural activity patterns: associative matrices that represent episodic memories may be instantiated in hippocampal synapses. The temporal context aligns with the hippocampus' recurrent nature, likely involving subregions including CA1, dentate gyrus [40], CA3 [41]), and cortical areas projecting to the hippocampus such as the entorhinal cortex [42].

Our results show strong connections to neural network models of episodic memory. For example, neural networks with attention [43] or recurrence [44] trained for free recall exhibit the same recall pattern as the optimal CMR. [45] showed that a neural network implementation of CMR can explain humans' flexible cognitive control. Our results also align with research connecting attention mechanisms in Transformers to the hippocampal formation [14]. While prior work focused on emergent place and grid cells in Transformers, the hippocampal subfields involved are also postulated to represent CMR components [40]. These results support our proposal that query-key-value attention mechanisms link to biological episodic retrieval, suggesting that CMR-like behavior emerges naturally in neural networks with proper objectives.

The link to induction heads could enable researchers to develop alternative mechanisms for episodic memory. For instance, K-composition and Q-composition induction circuits might serve as alternative models to CMR. K-composition and Q-composition require positional encoding, which we speculate could be implemented by grid-like cells with periodic activations tracking space and time [46]. Further, the interactions between episodic memory and other advanced cognitive functions [47–51] might be understood based on more complex attention-head composition mechanisms (e.g., N-th order virtual attention head [21]).

From the perspective of LLM mechanistic interpretability, we offer a more detailed behavioral description and reinterpret induction heads through the lens of CMR and the asymmetric contiguity bias. First, CMR-like heads are not limited to induction heads – some attention heads are well captured by CMR despite not meeting the traditional induction head criterion (e.g., Fig. 5d). Second, we speculate that heads with low CMR distances and low induction-head matching scores may encode multiple future tokens observed in LLMs [52], capturing distant token information better than ideal induction heads. Third, we observed a scale difference between raw attention scores and those from human recall, as discussed in Appendix G.1. Lastly, though CMR relies on recurrently updated context vectors that are different from the K-composition and Q-composition mechanisms, we posit that deeper Transformer models may develop similar mechanisms via autoregressively updated information in the residual stream, a possibility yet to be explored. Overall, our results suggest a fuller view of ICL mechanisms in LLMs, where heads learn attention biases akin to human episodic memory, empowering next-token and future-token predictions.

These connections with CMR may shed light on intriguing features and functions in LLMs. For instance, the "lost in the middle" phenomenon may be related to these heads [53], as humans exhibit similar recall patterns [19, 26]. Understanding the connection could suggest strategies to mitigate the problem, e.g., adjusting study schedules based on the serial position [54]. Second, CMR not only applies to individual words but also to clusters of items [55, 56], suggesting these heads may process hierarchically organized text chunks [57]. Third, episodic mechanisms captured by CMR are posited to support adaptive control [47–49] and flexible reinforcement learning [50, 51], suggesting similar roles for these heads in more complex cognitive functions of LLMs. Finally, it was proposed that Transformers can be implemented in a biologically plausible way [58], and we outline an alternative proposal mapping Transformer models to the brain (Appendix G.2).

Our study has several limitations. First, while we use CMR to characterize these heads' behavior, it is unclear if CMR can serve as a mechanistic model in larger Transformer models. Second, it is unclear if our conclusions hold for untested Transformer models. Third, our causal analysis lacks the power to narrow down the causal role of specific CMR-like characteristics. In addition, while ICL score is a primary metric for measuring ICL ability [15, 59], a systematic evaluation using specific datasets and tasks would allow for a stronger causal claim. Addressing these limitations is a key future direction.

## Acknowledgements

M.K.B was supported by NIH R01NS125298. M.K.B and L.J.-A. were supported by the Kavli Institute for Brain and Mind. The authors thank the Apart Lab for Interpretability Hackathon 3.0, and thank the support from Swarma Club and Post-ChatGPT Reading Group supported by the Save 2050 Programme jointly sponsored by Swarma Club and X-Order. In addition, the authors thank X. Li, H. Xiong, and the anonymous reviewers for their feedback.

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

## A  Code

All code is available at `https://github.com/corxyz/icl-cmr`.

## B  Attention mechanisms in Transformer models

For each attention head $h$, the standard self-attention mechanism calculates its output as $H^{(h)}(z_i^{(l-1)}; \{z_j^{(l-1)}\}) = W_O^{(h)} \sum_j W_V^{(h)} z_j^{(l-1)} A_{ij}^{(h)}$, where $W_O^{(h)}$ is the $d_{\text{model}} \times d_{\text{head}}$ output projection matrix, $W_V^{(h)}$ is the $d_{\text{head}} \times d_{\text{model}}$ value projection matrix, and $A_{ij}^{(h)}$ is the attention pattern attending from the query position $i$ to the key position $j$. The attention pattern matrix is $A^{(h)} = \text{Softmax}(\tilde{A}^{(h)}) = \text{Softmax}(z_j^{(l-1)\,T} W_Q^T W_K z_j^{(l-1)} / \sqrt{d_{\text{head}}})$, where $\tilde{A}^{(h)}$ represents the attention score matrix; $W_K^{(h)}$ and $W_Q^{(h)}$ are the $d_{\text{head}} \times d_{\text{model}}$ key and query projection matrices respectively.

## C  Detailed introduction of CMR

In this section, we offer a more detailed description of the Context Maintenance and Retrieval (CMR) model of human episodic memory, focusing on its core components and underlying dynamics. CMR operates in two distinct experimental phases: an encoding/learning phase, where the model memorizes a list of words, and a retrieval/recall phase, where it freely recalls the learned words. The CMR model consists of four primary components: the evolving temporal context ($\mathbf{t}$), the word representation or embedding ($\mathbf{f}$), the word-to-context association matrix ($\mathbf{M}^{\text{FT}}$), and the context-to-word association matrix ($\mathbf{M}^{\text{TF}}$).

The temporal context vector ($\mathbf{t}$) continuously evolves over time, incorporating recency-weighted historical information during both encoding and retrieval. The core dynamics of the temporal context $\mathbf{t}$ are governed by the update rule Eq. 1.

For illustrative purposes, in the rest of this section, we consider an example sequence consisting of five *distinct* one-hot word embeddings $\mathbf{f}_1, \mathbf{f}_2, ..., \mathbf{f}_5$. Each embedding is represented by a 6-dimensional vector, such that $\mathbf{f}_i$ contains a 1 at the $i$-th position, and 0 everywhere else. The 6-th dimension represents a dummy stimulus that only serves to maintain the unit norm of the context vector but otherwise irrelevant to the experiment. Additionally, we index encoding/learning time with $j$ and retrieval/recall time with $k$ to explicitly mark the two phases.

For simplicity, we set the pre-experimental word-to-context memory matrix $\mathbf{M}_{\text{pre}}^{\text{FT}}$ ($6 \times 6$) to the identity matrix. As the result of the distinctiveness assumption, each word embedding *at encoding* directly induces the same one-hot vector as the input context $\mathbf{t}_j^{\text{IN}} = \mathbf{M}_{\text{exp}}^{\text{FT}} \mathbf{f}_j = \mathbf{f}_j$. Let $\mathbf{t}_0 = (0, 0, 0, 0, 0, 1)^T$. Thus given a sequential presentation of $\mathbf{f}_1, \mathbf{f}_2, ..., \mathbf{f}_5$, the corresponding context vectors $\mathbf{t}_j$ are

$$\mathbf{t}_1 = (\beta, 0, 0, 0, 0, \rho)^T$$
$$\mathbf{t}_2 = (\rho\beta, \beta, 0, 0, 0, \rho^2)^T$$
$$\cdots$$
$$\mathbf{t}_5 = (\rho^4\beta, \rho^3\beta, \rho^2\beta, \rho\beta, \beta, \rho^5)^T,$$

where $\beta$ is the temporal drift parameter, and $\rho = \sqrt{1 - \beta^2}$ ensures that $\mathbf{t}_j$ has unit norm. Distinct parameters $\beta_{\text{enc}}$ and $\beta_{\text{rec}}$ are used by the encoding and retrieval phases respectively to capture the phase-specific temporal dynamics. Conceptually, the temporal context reflects a recency-weighted history of past information, with the parameter $\rho$ determining the rate of degradation over time.

More generally, the input context is determined by both pre-experimental and experimental contexts:

$$\mathbf{t}_j^{\text{IN}} = \mathbf{M}^{\text{FT}} \mathbf{f}_j = ((1 - \gamma_{\text{FT}}) \mathbf{M}_{\text{pre}}^{\text{FT}} + \gamma_{\text{FT}} \mathbf{M}_{\text{exp}}^{\text{FT}}) \mathbf{f}_j.$$

The memory matrix $\mathbf{M}^{\text{FT}}$ is therefore a weighted combination of pre-existing associations learned prior to encoding (the pre-experimental memory matrix $\mathbf{M}_{\text{pre}}^{\text{FT}}$) and associations learned during the encoding phase (the experimental memory matrix $\mathbf{M}_{\text{exp}}^{\text{FT}}$). $\mathbf{M}_{\text{pre}}^{\text{FT}}$ is commonly initialized as the

identity matrix at the beginning of each experiment and held constant throughout the experiment. On the other hand, $\mathbf{M}_{\text{exp}}^{\text{FT}}$ is usually initialized as the zero matrix. It is updated during the encoding phase at the presentation of each new word by associating the context vector $\mathbf{t}_{j-1}$ (value) with the embedding of the just encountered word $\mathbf{f}_j$ (key):

$$\mathbf{M}_{j,\text{exp}}^{\text{FT}} \leftarrow \mathbf{M}_{j-1,\text{exp}}^{\text{FT}} + \mathbf{t}_{j-1}\mathbf{f}_j^T$$

(Fig. 3c, first layer).

Suppose at the beginning of the experiment, the experimental word-to-context matrix $\mathbf{M}_{0,\text{exp}}^{\text{FT}}$ is initialized as the zero matrix. At encoding time $j = 1$, the experimental matrix $\mathbf{M}_{j,\text{exp}}^{\text{FT}}$ is updated by the pairwise association of the *previous* context and the current word:

$$\mathbf{M}_{1,\text{exp}}^{\text{FT}} = \mathbf{M}_{0,\text{exp}}^{\text{FT}} + \mathbf{t}_0\mathbf{f}_1^T$$

$$= \mathbf{M}_{0,\text{exp}}^{\text{FT}} + \begin{bmatrix} 0 \\ 0 \\ 0 \\ 0 \\ 0 \\ 1 \end{bmatrix} \begin{bmatrix} 1 & 0 & 0 & 0 & 0 & 0 \end{bmatrix}$$

$$= \begin{bmatrix} 0 & 0 & 0 & 0 & 0 & 0 \\ 0 & 0 & 0 & 0 & 0 & 0 \\ 0 & 0 & 0 & 0 & 0 & 0 \\ 0 & 0 & 0 & 0 & 0 & 0 \\ 0 & 0 & 0 & 0 & 0 & 0 \\ 1 & 0 & 0 & 0 & 0 & 0 \end{bmatrix}.$$

Similarly, at time $j = 2$, the experimental matrix incorporates the new context-word pair as

$$\mathbf{M}_{2,\text{exp}}^{\text{FT}} = \mathbf{M}_{1,\text{exp}}^{\text{FT}} + \mathbf{t}_1\mathbf{f}_2^T$$

$$= \mathbf{M}_{1,\text{exp}}^{\text{FT}} + \begin{bmatrix} \beta \\ 0 \\ 0 \\ 0 \\ 0 \\ \rho \end{bmatrix} \begin{bmatrix} 0 & 1 & 0 & 0 & 0 & 0 \end{bmatrix}$$

$$= \begin{bmatrix} 0 & \beta & 0 & 0 & 0 & 0 \\ 0 & 0 & 0 & 0 & 0 & 0 \\ 0 & 0 & 0 & 0 & 0 & 0 \\ 0 & 0 & 0 & 0 & 0 & 0 \\ 0 & 0 & 0 & 0 & 0 & 0 \\ 1 & \rho & 0 & 0 & 0 & 0 \end{bmatrix}.$$

The same update procedure continues until the end of the word sequence. Critically, note that the product of $\mathbf{M}_{j,\text{exp}}^{\text{FT}}$ and any $\mathbf{f}_{j+l}$ with $l > 0$ is the zero vector.

During retrieval, the most recently recalled word $\mathbf{f}_k$ (query) retrieves its corresponding input context

$$\mathbf{t}_k^{\text{IN}} = \mathbf{M}^{\text{FT}}\mathbf{f}_k$$
$$= (1 - \gamma_{\text{FT}})\mathbf{M}_{\text{pre}}^{\text{FT}}\mathbf{f}_k + \gamma_{\text{FT}}\mathbf{M}_{\text{exp}}^{\text{FT}}\mathbf{f}_k$$
$$= (1 - \gamma_{\text{FT}})\mathbf{M}_{\text{pre}}^{\text{FT}}\mathbf{f}_k + \gamma_{\text{FT}} \sum_l (\mathbf{t}_{l-1}\mathbf{f}_l^T)\mathbf{f}_k$$
$$= (1 - \gamma_{\text{FT}})\mathbf{M}_{\text{pre}}^{\text{FT}}\mathbf{f}_k + \gamma_{\text{FT}} \sum_l \mathbf{t}_{l-1}(\mathbf{f}_l^T\mathbf{f}_k).$$

Let $\mathbf{f}_j$ denote the word encountered at encoding time $j$ with $j < k$. Assuming $\mathbf{f}_k = \mathbf{f}_j$ (i.e., the most recent recall $\mathbf{f}_k$ matches a previously encoded word $\mathbf{f}_j$), we have

$$\mathbf{t}_k^{\text{IN}} = (1 - \gamma_{\text{FT}})\mathbf{M}_{\text{pre}}^{\text{FT}}\mathbf{f}_j + \gamma_{\text{FT}}\mathbf{t}_{j-1}.$$

Furthermore, because we assumed $\mathbf{M}^{\mathrm{FT}}_{\mathrm{pre}}$ to be the identity matrix,

$$\mathbf{t}^{\mathrm{IN}}_k = (1 - \gamma_{\mathrm{FT}})\mathbf{f}_j + \gamma_{\mathrm{FT}}\mathbf{t}_{j-1}. \tag{S1}$$

This input context is subsequently reinstated along with the current context $\mathbf{t}_{k-1}$ at (retrieval) time $k$, resulting in an updated context $\mathbf{t}_k$ via Eq. 1. Thus $\mathbf{t}_k$ also incorporates information from $\mathbf{f}_j$ (the previous instance of the most recent recall) and $\mathbf{t}_{j-1}$ (the context immediately before presentation of the previous instance).

Importantly, the context $\mathbf{t}_j$ at encoding is also a linear combination of $\mathbf{f}_j$ and $\mathbf{t}_{j-1}$ (by Eq. 1),

$$\mathbf{t}_j = \beta\mathbf{t}^{\mathrm{IN}}_j + \rho\mathbf{t}_{j-1},$$

where $\mathbf{t}^{\mathrm{IN}}_j = ((1 - \gamma_{\mathrm{FT}})\mathbf{M}^{\mathrm{FT}}_{\mathrm{pre}} + \gamma_{\mathrm{FT}}\mathbf{M}^{\mathrm{FT}}_{\mathrm{exp}})\mathbf{f}_j$. However, because all word embeddings are one-hot and distinct, $\mathbf{M}^{\mathrm{FT}}_{\mathrm{exp}}$ has *not* learned any associations involving the word $\mathbf{f}_j$ prior to time $j$. Therefore, the second term is zero, and

$$\mathbf{t}_j = \beta\mathbf{f}_j + \rho\mathbf{t}_{j-1}. \tag{S2}$$

Comparing Eq. S2 and Eq. S1 thus reveals that $\mathbf{t}^{\mathrm{IN}}_k$, thus $\mathbf{t}_k$, become most similar to $\mathbf{t}_j$.

Finally, the context-to-word association matrix $\mathbf{M}^{\mathrm{TF}}$ is responsible for retrieving the context associated with a given word. $\mathbf{M}^{\mathrm{TF}}$ is usually initialized as the zero matrix prior to experiments. During encoding, it is updated to associate each word embedding $\mathbf{f}_j$ (value) with the corresponding context vector $\mathbf{t}_{j-1}$ (key):

$$\mathbf{M}^{\mathrm{TF}}_j \leftarrow \mathbf{M}^{\mathrm{TF}}_{j-1} + \mathbf{f}_j\mathbf{t}^T_{j-1}$$

(Fig. 3, second layer). In practice, we chose to maintain $\mathbf{M}^{\mathrm{TF}}$ as the transpose of $\mathbf{M}^{\mathrm{FT}}_{\mathrm{exp}}$. Because the updated context $\mathbf{t}_k$ is most similar to $\mathbf{t}_j$ (measured by their inner product), the retrieved vector

$$\begin{aligned}
\mathbf{f}^{\mathrm{IN}}_{k+1} &= \mathbf{M}^{\mathrm{TF}}\mathbf{t}_k \\
&= \sum_l (\mathbf{f}_l\mathbf{t}^T_{l-1})\mathbf{t}_k \\
&= \sum_l \mathbf{f}_l(\mathbf{t}^T_{l-1}\mathbf{t}_k)
\end{aligned}$$

will be the most similar to the word embedding $\mathbf{f}_{j+1}$. Thus, CMR will most likely recall $\mathbf{f}_{j+1}$ following a recall of $\mathbf{f}_k = \mathbf{f}_j$. In the main text, we specified $i = j + 1$.

Consequently, the CMR model explains two key phenomena observed in human free recall experiments: the temporal contiguity effect and the forward asymmetry effect. The temporal contiguity effect arises because (i) $\mathbf{t}_k$ integrates the information of $\mathbf{t}_{j-1}$, and (ii) the context vector $\mathbf{t}$ drifts over time, making $\mathbf{t}_{j-1}$ more similar to context vectors associated with nearby words in the sequence, favoring the retrieval of words close to $\mathbf{f}_j$. In addition, the forward asymmetry effect occurs because (i) $\mathbf{t}_k$ integrates the information of $\mathbf{t}^{\mathrm{IN}}_j$, and (ii) $\mathbf{t}^{\mathrm{IN}}_j$ contributes to $\mathbf{t}_j = \mathbf{t}_{i-1}$ and its following context vectors, but does not affect earlier context vectors. As a result, $\mathbf{t}_k$ preferentially retrieves words that follow $\mathbf{f}_j$, rather than those that precede it.

## D  Prompt design

We used the prompt of repeated random tokens due to several reasons (1) it aligns with human free recall experiments, where random words are presented sequentially [18]; (2) it is a widely acknowledged definition of induction heads in mechanistic interpretability literature [21, 15, 31, 60]; (3) it uses off-distribution prompts to focus on abstract properties, avoiding potential confounds from normal token statistics [21].

The prompt is constructed by taking the top N=100 most common English tokens with a leading space (to avoid unwanted tokenization behavior), which are tokens with the largest biases in the unembedding layer of GPT2-small (or the Pythia models). The prompt concatenated two copies of the permuted word sequence and had a total length of $2N + 1$ (one end-of-sequence token at the beginning). The two copies correspond to the study (encoding) and recall phases, respectively.

# E    Metrics & definitions

## E.1    Metrics for induction heads

Formally, we define the induction-head target pattern (i.e., attention probability distribution of an ideal induction head) $\bar{\mathbf{A}} = (\bar{a}_{d,s})$ over a sequence of tokens $\{x_i\}$ as

$$\bar{a}_{d,s} = \begin{cases} 1, & \text{if } x_{s-1} = x_d \text{ and } s < d \\ 0, & \text{otherwise} \end{cases}.$$

Here, $d$ is the destination position and $s$ is the source position. We then assess the extent to which each attention head performs this kind of prefix matching [15, 25]. Specifically, the induction-head matching score for a head with attention pattern $\mathbf{A} = (a_{d,s})$ is defined as

$$(\sum_d \sum_s a_{d,s} \bar{a}_{d,s}) / (\sum_d \sum_s a_{d,s}) \in [0, 1]$$

(Fig. 2a-b). A head that always performs ideal prefix matching will have an induction-head matching score of 1.

Additionally, an induction head should write to the current residual stream to increase the corresponding logit of the attended token (token copying). We adopt the copying score [21] to measure each head's tendency of token copying. In particular, consider the circuit $W = W_U W_O W_V W_E$ for one head, where $W_E$ defines token embeddings, $W_V$ computes the value of each token from the residual stream (i.e., aggregated outputs from all earlier layers), $W_O$ computes the head's output using a linear combination of the value vectors and the head's attention pattern, and $W_U$ unembeds the output to predict the next token. The copying score of the head is equal to

$$(\sum_k \lambda_k) / (\sum_k |\lambda_k|) \in [-1, 1],$$

where $\lambda_k$'s are the eigenvalues of the matrix $W$. Since copying requires positive eigenvalues (corresponding to increased logits), an induction head should have a positive copying score. An ideal induction head will have a copying score of 1.

## E.2    CRP analysis of attention heads

For a head with attention scores $\tilde{\mathbf{A}}$, the average attention score $\tilde{\alpha}_{\text{lag}}$ is defined as

$$\tilde{\alpha}_{\text{lag}} = \frac{1}{N - |\text{lag}| * 2} \sum_{|\text{lag}| < s \leq N - |\text{lag}|} \tilde{a}_{s+N, s+\text{lag}},$$

where $N$ is the length of the first repeat in the prompt. Thus if $\text{lag} = 0$, $\tilde{\alpha}_{\text{lag}}$ quantifies how much the first instance of a token is attended to on average; for $\text{lag} = 1$, $\tilde{\alpha}_{\text{lag}}$ quantifies the average amount of attention distributed to the immediately following token in the first repeat. We used $\text{lag} \in [-5, 5]$ throughout the paper.

## E.3    CMR distance

The metric *CMR distance* is defined as

$$d_{\text{CMR}} = \min_{\mathbf{q}} \left\{ \sum_{\text{lag}} \frac{(q_{\text{lag}} - \tilde{\alpha}_{\text{lag}})^2 / \text{Var}\,(\tilde{\alpha}_{\text{lag}})}{N_{\text{lag}}} \right\}.$$

Here, $N_{\text{lag}}$ is the number of distinct lags, and $q_{\text{lag}}$ is obtained by calculating the CRP using CMR with specific parameter values. Note we did not consider the full set of possible $\mathbf{q}$ but only the subset given by the combinations of parameters $\beta_{\text{enc}} = 0.05, 0.1, \ldots, 1$, $\beta_{\text{rec}} = 0, 0.05, \ldots, 1$, and $\gamma_{\text{FT}} = 0, 0.1, \ldots, 1$ for model fitting.

### E.4 Gaussian distance

The metric *Gaussian distance*, serving as a baseline, is defined as

$$d_{\text{Gaussian}} = \min_{\mathbf{g}} \left\{ \sum_{\text{lag}} \frac{(g_{\text{lag}} - \tilde{\alpha}_{\text{lag}})^2 / \text{Var}(\tilde{\alpha}_{\text{lag}})}{N_{\text{lag}}} \right\},$$

where $g_{\text{lag}}$ is the Gaussian function, defined as

$$g_{\text{lag}} = c_1 \exp\left\{ -\frac{(\text{lag} - c_2)^2}{2c_3^2} \right\} + c_4,$$

and $c_1$, $c_2$, $c_3$, $c_4$ are coefficients.

### E.5 Choosing thresholds for CMR distance

To identify heads with CMR-like attention scores and exclude non-CMR-like heads, we operationally chose a threshold of 0.5 in the main text. This threshold is informed by both the quantitative distribution of individual head CMR distances and exploratory qualitative analysis of the attention patterns. First, as the bottom panel in Fig. 5e shows, the distribution of individual head CMR distances in GPT2 can be divided roughly into two clusters: a cluster peaking around 0.1 (and extending to around 1), and a spread-out cluster around 2.4. Second, we examined heads with 1 CMR distance and found non-human-like CRP (e.g., Fig. S2a,b). Further, many heads with CMR distance around 0.6-0.8 again show CRP different from humans (e.g., Fig. S2c, d). Finally, we also tested a threshold of 0.1 and found no qualitative difference (Fig. Fig. S3c, d).

# F  Comparison of composition mechanisms of induction heads and CMR

| | | K-composition | Q-composition | CMR |
|---|---|---|---|---|
| Representation (residual stream) before $H_1$ | | $PE_i$ & $TE_i$ [At $i$]
$PE_j$ & $TE_j$ [At $j$]
$PE_k$ & $TE_k$ [At $k$] | $PE_i$ & $TE_i$ [At $i$]
$PE_j$ & $TE_j$ [At $j$]
$PE_k$ & $TE_k$ [At $k$] | $t_{i-1}$ & $f_i$ [At $i$]
$t_{j-1}$ & $f_j$ [At $j$]
$t_{k-1}$ & $f_k$ [At $k$] |
| First-layer head $H_1$ | Type | previous token head | duplicate token head | word-to-context matrix $M^{\mathrm{FT}}$ |
| | Query | $PE_i$ [At $i$] | $TE_k$ [At $k$] | $f_k$ [At $k$] |
| | Key | $PE_j$ [At $j$] | $TE_j$ [At $j$] | $f_j$ [At $j$] |
| | Optimal Q-K match condition | $PE_{j+1} = PE_i$ | $TE_j = TE_k$ | $f_j = f_k$ |
| | Activation | Softmax | Softmax | Linear |
| | Value | $TE_j$ [At $j$] | $PE_j$ [At $j$] | $t_{j-1}$ [At $j$] |
| | Output | $TE_j$ [At $i$]* | $PE_j$ [At $k$]* | $t_{j-1}$ [At $k$] |
| Representation (residual stream) after $H_1$ | | $PE_i$ & $TE_i$ & $TE_j$ [At $i$]
$PE_j$ & $TE_j$ [At $j$]
$PE_k$ & $TE_k$ [At $k$] | $PE_i$ & $TE_i$ [At $i$]
$PE_j$ & $TE_j$ [At $j$]
$PE_k$ & $TE_k$ & $PE_j$ [At $k$] | $t_{i-1}$ & $t_i$ & $f_i$ [At $i$]
$t_{j-1}$ & $t_j$ & $f_j$ [At $j$]
$t_{k-1}$ & $t_k$ & $f_k$ [At $k$] † |
| second-layer head $H_2$ | Type | induction head | induction head | context-to-word matrix $M^{\mathrm{TF}}$ |
| | Query | $TE_k$ [At $k$] | $PE_j$ [At $k$] | $t_j$ [At $k$]† |
| | Key | $TE_j$ [At $i$] | $PE_i$ [At $i$] | $t_{i-1}$ [At $i$] |
| | Optimal Q-K match condition | $TE_j = TE_k$ | $PE_{j+1} = PE_i$ †† | $t_j = t_{i-1}$ †† |
| | Activation | Softmax | Softmax | Linear |
| | Value | $TE_i$ [At $i$] | $TE_i$ [At $i$] | $f_i$ [At $i$] |
| | Output | $TE_i$ [At $k$]* | $TE_i$ [At $k$]* | $f_{k+1}^{\mathrm{IN}}$ [At $k$] |
| Representation (residual stream) after $H_2$ | | $PE_i$ & $TE_i$ & $TE_j$ [At $i$]
$PE_j$ & $TE_j$ [At $j$]
$PE_k$ & $TE_k$ & $TE_i$ [At $k$] | $PE_i$ & $TE_i$ [At $i$]
$PE_j$ & $TE_j$ [At $j$]
$PE_k$ & $TE_k$
& $PE_j$ & $TE_i$ [At $k$] | $t_{i-1}$ & $t_i$ & $f_i$ [At $i$]
$t_{j-1}$ & $t_j$ & $f_j$ [At $j$]
$t_{k-1}$ & $t_k$
& $f_k$ & $f_{k+1}^{\mathrm{IN}}$ [At $k$] |

Table S1: **Comparison of mechanisms of induction heads and CMR.**
$PE_i$: position embedding for the token at index/position $i$.
$TE_i$: token embedding for the token at index/position $i$.
$t_{i-1}$: context vector associated with the word at index/position $i$.
$f_i$: word embedding for the word at index/position $i$.
[At $i$]: information available at index/position $i$.
*: The output is approximate (due to the weighted-average over all previous tokens).
†: $t_k$ is updated from $t_{k-1}$ by $t_{j-1}$ (due to $M_{\mathrm{exp}}^{\mathrm{FT}}$) and $t_j^{\mathrm{IN}}$ (due to $M_{\mathrm{pre}}^{\mathrm{FT}}$), thus containing the information of $t_j$ (updated from $t_{j-1}$ by $t_j^{\mathrm{IN}}$).
††: The optimal Q-K match condition in Q-composition requires transformation from $PE_j$ to $PE_{j+1}$, implemented by the $W_Q W_K$ matrix of $H_2$. The optimal Q-K match condition in CMR requires transformation from $t_{j-1}$ to $t_j$, implemented by $t_j^{\mathrm{IN}}$.

# G  Supplementary discussion

## G.1  Scale difference

We have observed the scale difference between raw attention scores and those from human recall. We can speculate a few reasons. (i) It's common to tune the temperature only at the last readout layer, and it's unclear if Transformers benefit from variable temperatures in other attention layers. (ii) The range of attention scores might depend on the distributions of the model's training and evaluation data (e.g., evaluation prompts). (iii) Unlike LLMs, humans may not fully optimize next-word predictions, as biological objectives are more complex and varied. Recall might constitute only one use case, with the cognitive infrastructure also engaged in tasks like spatial navigation and adaptive decision-making. The more moderate scale in humans may thus reflect tradeoffs between multiple objectives.

## G.2 Mapping components in Transformer models to the brain

Here, we propose that the MLP layers might be mapped to the cortex and the self-attention layers might be mapped to the hippocampus. The model parameters could be encoded by slowly updated synapses in the cortex and hippocampus, with the key-value associations stored in fast Hebbian-like hippocampal synapses. The residual stream updated by MLP and attention layers may be akin to the activation-based working memory quickly updated by the cortico-hippocampal circuits.

# H Experiment compute resources

The induction-head matching scores and copying scores of each head in GPT2-small and all Pythia models are computed using Google Colab Notebook. All models were pretrained and accessible through the TransformerLens library [25] with MIT License and used as is. See Table S2 for details.

| Transformer Model | Type of compute worker | RAM (GB) | Storage (GB) | Computing time (minutes) |
|---|---|---|---|---|
| GPT2-small | CPU | 12.7 | 225.8 | < 1 |
| Pythia-70m-deduped-v0 | CPU | 12.7 | 225.8 | 2 |
| Pythia-160m-deduped-v0 | CPU | 12.7 | 225.8 | 5 |
| Pythia-410m-deduped-v0 | CPU | 12.7 | 225.8 | 15 |
| Pythia-1b-deduped-v0 | CPU | 12.7 | 225.8 | 45 |
| Pythia-1.4b-deduped-v0 | High-RAM CPU | 51.0 | 225.8 | 56 |
| Pythia-2.8b-deduped-v0 | High-RAM CPU | 51.0 | 225.8 | 161 |
| Pythia-6.9b-deduped-v0 | TPU v2 | 334.6 | 225.3 | 111 |
| Pythia-12b-deduped-v0 | TPU v2 | 334.6 | 225.3 | 205 |
| Qwen-7b | TPU v2 | 334.6 | 225.3 | 7 |
| Mistral-7b | TPU v2 | 334.6 | 225.3 | 5 |
| Llama3-8b | TPU v2 | 334.6 | 225.3 | 10 |

Table S2: **Details of compute resources used to compute induction head metrics**. All models were pretrained and accessible through the TransformerLens library [25] with MIT License. The numbers in the "Computing time" column indicate the total number of minutes it took to compute all scores for all heads across all checkpoints where available.

We used an internal cluster to compute the subset of $\mathbf{q}$ (CRP) for model fitting. The internal cluster has 6 nodes with Dual Xeon E5-2699v3, which has 72 threads and 256GB RAM per thread, plus 4 nodes with Dual Xeon E5-2699v3, which has 72 threads and 512GB RAM per thread. This computation took a total of 90 hours.

Finally, model fitting was done on a 2023 MacBook Pro with 16GB RAM. All experiments were completed within 1 hour regardless of model size. This section contains all experiments we conducted that required non-trivial computational resources.

# I  Additional figures

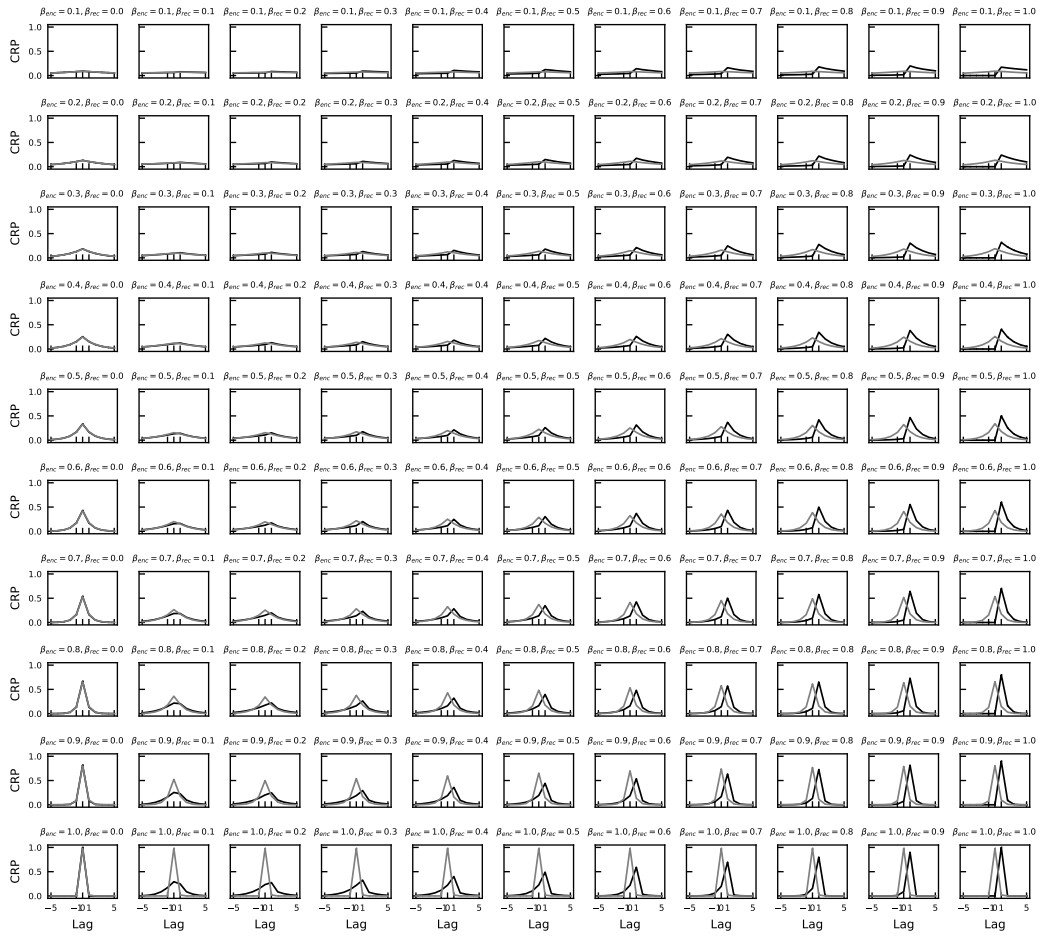

Figure S1: **Example conditional response probability distributions produced by different parameterizations of CMR.** Black lines correspond to $\gamma_{\mathrm{FT}} = 0$ (i.e., only the pre-experimental contexts are used during retrieval). Grey lines correspond to $\gamma_{\mathrm{FT}} = 1$ (i.e., only the experimental contexts are used during retrieval).

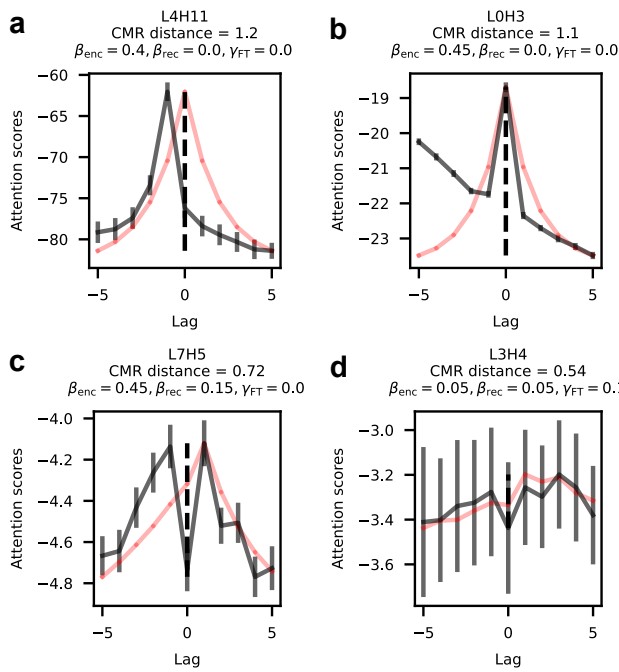

Figure S2: Example GPT2 attention heads with a CMR distance roughly between 0.5 and 1.0.

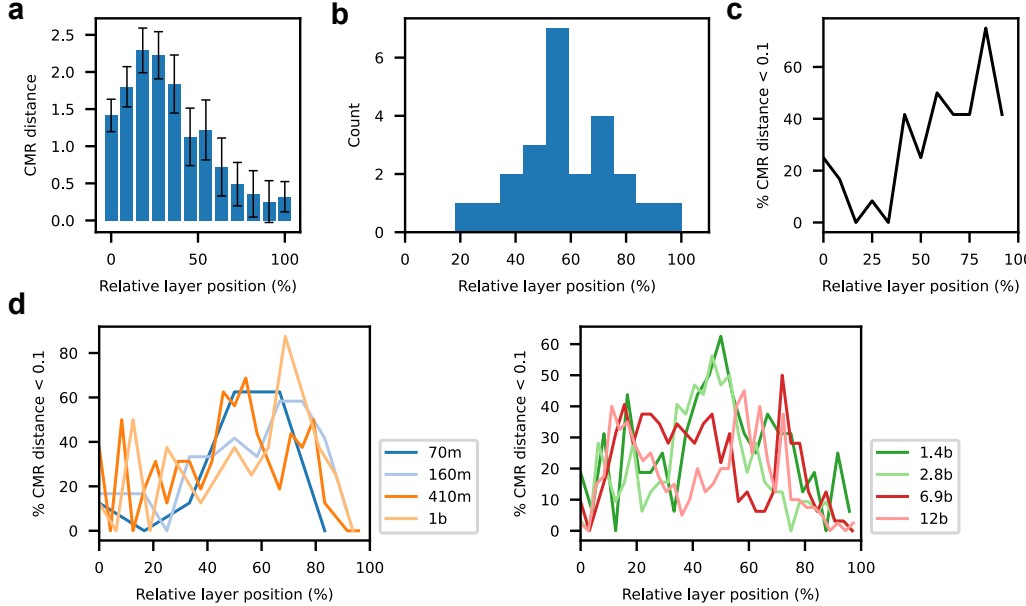

Figure S3: **CMR distances vary with relative layer positions in LLMs. (a)** Average CMR distance across heads in each layer of GPT2-small. Error bars indicate standard errors. **(b)** Histogram of the relative positions of the two layers with the lowest average CMR distances in each of the following models: GPT2-small, all Pythia models, Qwen-7B, Mistral-7B, and Llama3-8B. **(c)** Percentage of heads with a CMR distance less than 0.1 in GPT2-small. **(d)** Same as in **(c)** except for Pythia models.

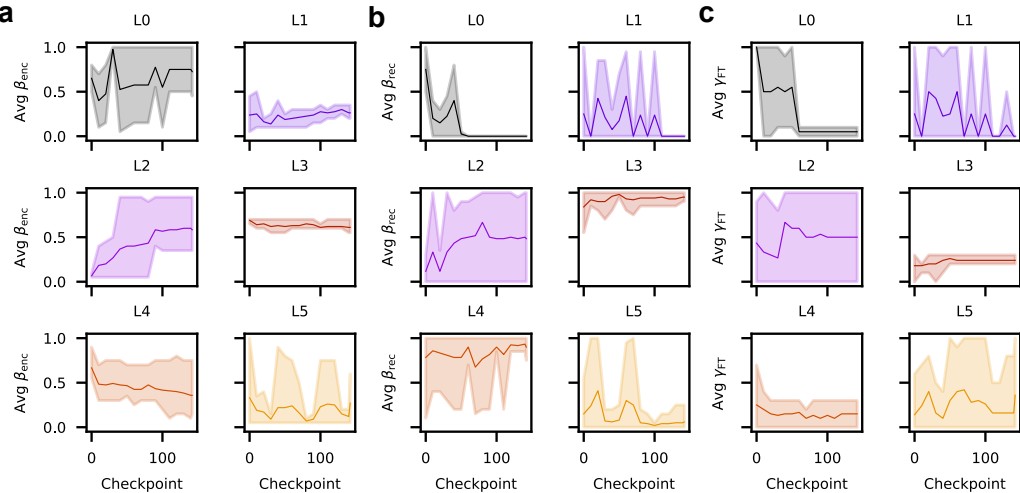

Figure S4: **Fitted CMR parameters of Pythia-70m model over training.** Solid lines indicate values averaged across heads with a CMR distance lower than 0.5 in the corresponding layer. Shaded areas indicate the range of fitted values (the lower edge indicates the minimum value; the upper edge indicates the maximum value). **(a-b)** Fitted CMR temporal drift parameters $\beta_{\mathrm{enc}}$ (a), and $\beta_{\mathrm{rec}}$ (b) in different layers as a function of training time. **(c)** Fitted experimental context mix parameter $\gamma_{\mathrm{FT}}$ in different layers as a function of training time.

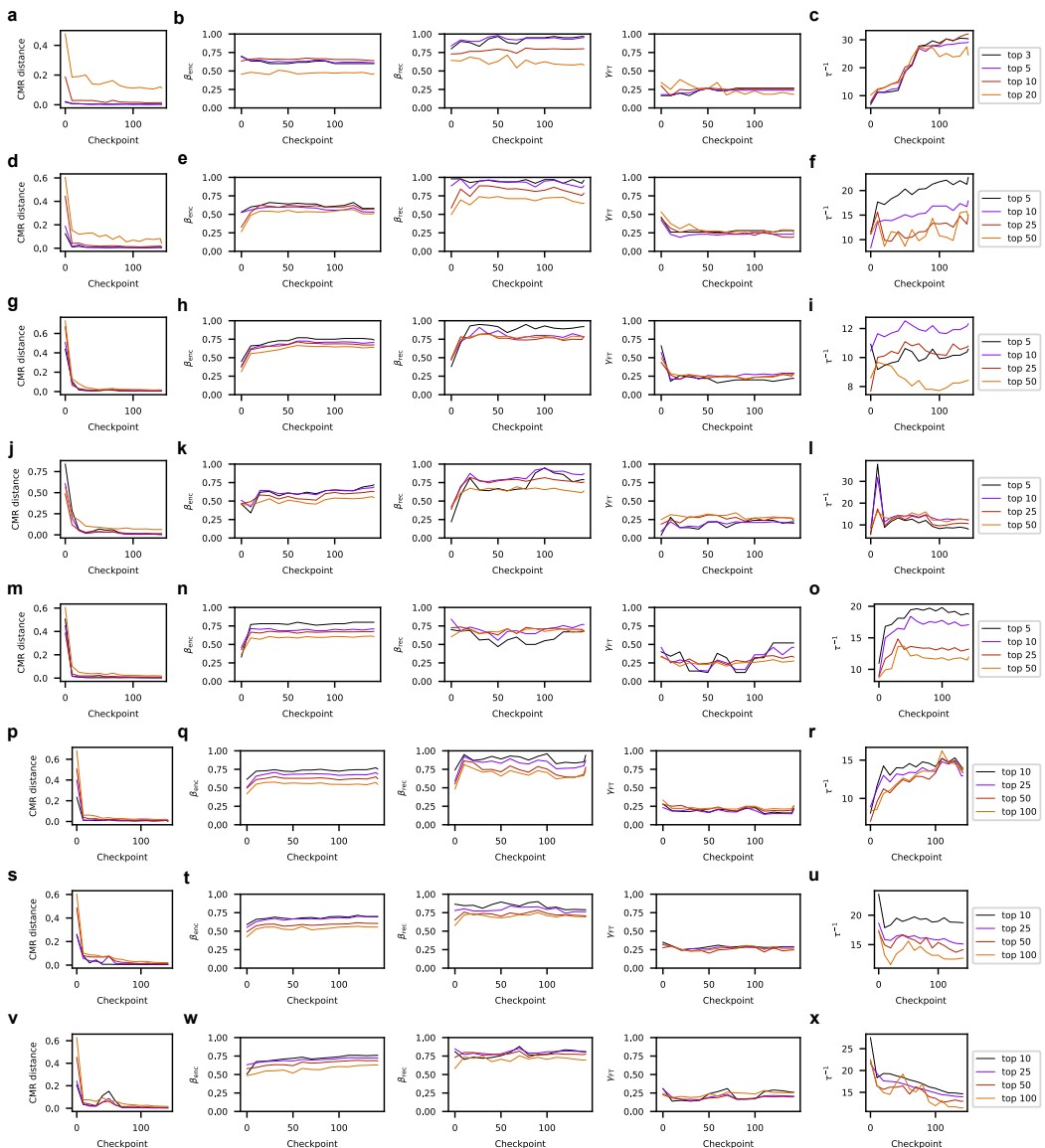

Figure S5: **Behavior of top attention heads in different Pythia models. (a)** CMR distance of the top CMR-like heads in Pythia-70m as a function of training time. Heads are selected by ranking all attention heads by their CMR distances (e.g., "top 3" heads correspond to the three lowest CMR distances). **(b)** Fitted CMR temporal drift parameters $\beta_{\mathrm{enc}}$(left), $\beta_{\mathrm{rec}}$ (middle), $\gamma_{\mathrm{FT}}$ (right) of the top CMR-like heads in Pythia-70m as a function of training time. **(c)** Average fitted inverse temperature of the top CMR-like heads in Pythia-70m as a function of training time. **(d-x)** Same as a-c except for Pythia-160m (d-f), Pythia-410m (g-i), Pythia-1b (j-l), Pythia-1.4b (m-o), Pythia-2.8b (p-r), Pythia-6.9b (s-u), and Pythia-12b (v-x).

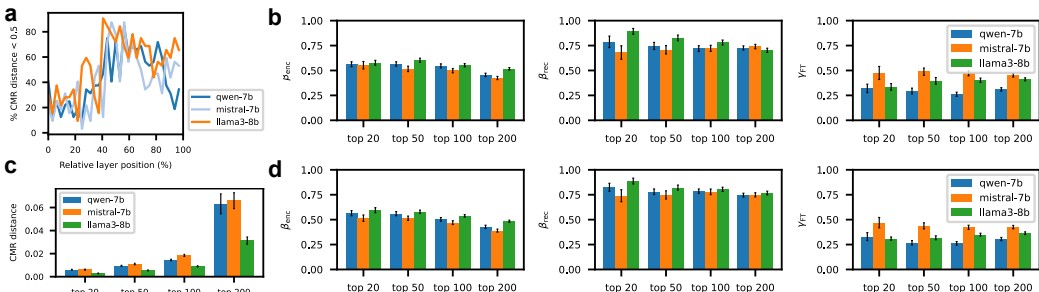

Figure S6: **Generalizing results to additional models.** **(a)** Percentage of heads with a CMR distance less than 0.5 in different layers. **(b)** Fitted CMR temporal drift parameters $\beta_{\mathrm{enc}}$(left), $\beta_{\mathrm{rec}}$ (middle), $\gamma_{\mathrm{FT}}$ (right) in attention heads with the highest induction-head matching scores (i.e., top induction heads). **(c)** CMR distance of top induction heads in different models. **(d)** Same as **(b)** except for top CMR-like heads (i.e., attention heads with the lowest CMR distances).

