# OpenReview forum: "Linking In-context Learning in Transformers to Human Episodic Memory"
_NeurIPS.cc/2024/Conference — NeurIPS 2024 poster_

### Official Review · Reviewer_18Dm · 2024-07-06

**Soundness:** 3
**Presentation:** 4
**Contribution:** 3
**Rating:** 6
**Confidence:** 4

**Summary:**

This work establishes a correspondence between 1) induction heads, known to functionally contribute to LLM's in-context learning, and 2) the CMR model of human episodic memory. They showed mechanistic equivalences between the model components, relating the Q-composition realization of induction heads to CMR. They also replicated an analysis of human episodic recall over different temporal token position distances on the attention scores of the induction heads in pre-trained LLMs and found human-like patterns.

**Strengths:**

The paper establishes a nice link between transformers and human memory mechanisms, which helps the field think about the role of induction heads as how episodic memory mechanisms may contribute to language generation, especially from the angle of drifting temporal context. The normative interpretation is very interesting.

The overall presentation is clear and easy to follow.

**Weaknesses:**

The stated missing connection between transformer and neuroscience seems overstated given prior work relating the attention mechanism to the Hopfield network and transformers with hippocampus representations.

I'm uncertain how informative the CMR distance metric is since CMR seems to flexibly fit to many curves. A strong link to human episodic memory would require the specific characteristics of temporal contiguity and forward asymmetry to be present in a given head.

The authors make a nice connection, but it would be great if the paper can dive deeper into why this may be a meaningful connection. For example, considering that episodic memory is about context-dependent retrieval of prior-related information, but ICL can lead to some degree of generalization to novel tasks.

**Questions:**

Since the model CRP curve uses pre-softmax attention scores, the actual transition probabilities would be of a very different scale with human recall CPR. I'm curious how the authors interpret such a scale difference.

The beginning of section 5.2 seems to suggest some fine-tuning for the target task, which was a surprise as I thought the model evaluation was purely in-context prompting. Could the authors clarify?

Would K-composition still show the same CRP-like patterns? Is there a way to differentiate the two, e.g. by empirical measures or by creating hard-wired 2-layer models?

In Fig5, many GPT2 heads have really small CMR distance but don’t have high induction head matching scores. What could the functional role of these other low CMR-distance heads be?

**Limitations:**

Stated in the paper.

---

> ### Author Rebuttal · Authors · 2024-08-07
>
> Thank you for your appreciation and insightful feedback. Below is our point-by-point response:
>
> ### Weaknesses
> 1. You’re right that prior works have linked Transformers to neuroscience. In the introduction, we referenced Whittington et al. (2022) on the relationship between attention mechanisms and hippocampal place cells. While the attention mechanism is equivalent to the update rule in continuous-state Hopfield networks (Ramsauer et al., 2021), its direct connection to neuroscience or the brain is less clear. Generally it’s not surprising to find connections between Transformers and memory, as the attention mechanism directly generalizes the old idea of key-value memory networks. However, our focus is on emergent capabilities rather than explicitly designed model components, particularly the emergent behavior of interacting heads rather than individual ones. We will revise the introduction to specify this gap and avoid overstatement.
> 2. Your point about connecting human episodic memory to fitting CMR to heads is well-taken. CMR is designed to model episodic recall patterns, especially the asymmetric contiguity bias. It only captures behavior with temporal contiguity and forward asymmetry (or complete symmetry); no CMR parameterization would produce a CRP curve that’s temporally discontiguous or backwardly asymmetric. A good fit (low CMR distance) usually requires temporal contiguity and forward asymmetry, as observed in human subjects and attention heads. We provide a direct comparison with human data in global rebuttal point 2 and examples in Fig. R3, where high CMR distances reflect temporal discontiguity and/or backward asymmetry. We also did an ablation study suggesting a causal link between low CMR distances and a model’s ICL ability (see global rebuttal point 4).
> 3. This is a great observation, which we are exploring in other projects. Existing works suggest that context-dependent episodic mechanisms support adaptive control and flexible reinforcement learning, including generalization to novel decision and categorization tasks (e.g., Kumaran & McClelland, 2012; Gershman & Daw, 2017; Zhou et al., 2023; Giallanza et al., 2024). By linking context-driven episodic retrieval and ICL, we aim to reveal normative and mechanistic explanations of generalizable knowledge that are universal in humans and machines.
>
> ### Questions
> 1. Thank you for the insightful question. We’ve noticed the scale difference between raw attention scores and those from human recall. We can speculate a few reasons. (i) It’s common to tune the temperature only at the last layer, so it’s unclear if Transformers benefit from variable temperatures in other layers. (ii) The range of attention scores might depend on the distributions of the model’s training and evaluation data (e.g., evaluation prompts). (iii) Unlike LLMs, humans may not fully optimize next-word predictions, as biological objectives are more complex and varied. Recall might constitute only one use case, with the cognitive infrastructure also engaged in tasks like spatial navigation and adaptive decision-making. The more moderate scale in humans may thus reflect tradeoffs between multiple objectives.
> 2. Thank you for the opportunity to clarify our claim. Section 5.2 states, “As the model's loss on the designed prompt decreases through training (Fig. 7a), the degree of temporal clustering increases, especially in layers where induction heads usually emerge.” You are right that we didn’t fine-tune for the target task and only evaluated the model with in-context prompting. We took different checkpoints of pre-trained models as-is, computed their losses on the designed prompt, and assessed their in-context capability and degree of temporal clustering at different training stages. This analysis helps identify when induction/CMR-like heads and ICL abilities emerge during training.
> 3. We appreciate your thoughtful question. Based on existing findings (e.g., Elhage et al., 2021), we doubt that attention patterns between K- and Q-composition would differ qualitatively in hard-wired 2-layer Transformers or larger models—they likely both exhibit CMR-like CRP patterns. Since ICL in large models may arise from more complex head-composition mechanisms (noted in Olsson et al., 2022), empirically identifying attention pattern differences will be challenging. The only reliable way to distinguish between mechanisms is to examine head interactions via attention patterns, outputs, and weights ($W_{QK}$, $W_{OV}$, etc.).
> 4. Thank you for noting this. For example, the head in Fig. 5d exhibits the highest attention on the previous occurrence of the current token and can be categorized as a duplicate token head (Wang et al., 2022). Transformer models trained to predict the next token also learn to encode several tokens in the future (Pal et al., 2023). We speculate that heads with low CMR distances and low induction-head matching scores might play a role here, as they encode distant token information more than an ideal induction head. Understanding these CMR-like heads' functional roles remains an open question.
>
> **References**
> - Ramsauer et al. (2021). Hopfield Networks is All You Need.
> - Kumaran & McClelland. (2012). Generalization Through the Recurrent Interaction of Episodic Memories.
> - Gershman & Daw. (2017). Reinforcement Learning and Episodic Memory in Humans and Animals: An Integrative Framework.
> - Zhou et al. (2023). Episodic retrieval for model-based evaluation in sequential decision tasks.
> - Giallanza et al. (2024) Toward the Emergence of Intelligent Control: Episodic Generalization and Optimization.
> - Elhage et al. (2021). A mathematical framework for transformer circuits.
> - Olsson et al. (2022) In-context Learning and Induction Heads.
> - Pal et al. (2023). Future Lens: Anticipating Subsequent Tokens from a Single Hidden State.
> - Wang et al. (2022). Interpretability in the wild: a circuit for indirect object identification in gpt-2 small.

---

> > ### Comment · Reviewer_18Dm · 2024-08-11
> >
> > Thank you for these detailed responses. I appreciate the effort going into the additional experiments on different models and datasets. I do think that the causal claim should be handled carefully. The ablation shows that low-CMR-distance heads are more important for ICL vs. random heads, but it doesn't directly transfer to a conceptual causal link about the removal of episodic-memory-like characteristics in the model to be responsible for the drop in ICL abilities, as it could be something else about these heads and their interaction with other heads (especially given that the ICL score used is only a heuristic metric; though I am not familiar to what extent this particular score is used to approximate ICL abilities in the interpretability field. I imagine ideally we would want a set of datasets/tasks to derive a performance-based ICL metric). In general, as the authors mentioned in the response and are exploring in other projects, I think if the authors can contextualize the results in more discussions on what and why episodic-memory-like mechanisms may be key to language modeling and ICL, it would enhance the paper further.

---

> > > ### Author Response · Authors · 2024-08-14
> > >
> > > Thanks for acknowledging our effort. We will discuss contextualization of episodic memory mechanisms in language modeling and ICL, e.g., on relating episodic memory to flexible language processing in humans (Duff & Brown-Schmidt, 2012).
> > >
> > > We will include the discussion of implications and limitations of our causal analysis using the ICL score. We agree that our results demonstrate the causal importance of CMR-like heads, while it would be more challenging to narrow down the causal role of CMR-like characteristics (e.g., ablating specific characteristics but keeping all other intact). More generally, the ablation of selected heads remains the primary method in the MI community to test a causal claim about the characteristics of attention heads on model functions (Nanda et al., 2023; Olsson et al., 2022; Wang et al., 2022; Meng et al., 2023; Chan et al., 2022). Additionally, we note that the ICL score is one of the main metrics that the MI community uses to measure the ICL ability (Lee et al., 2024; Olsson et al., 2022), but we agree that a systematic evaluation of ICL ability using multiple datasets and tasks will provide stronger evidence. These limitations and considerations will be explicitly addressed in our discussion section.

---

### Official Review · Reviewer_uMwU · 2024-07-10

**Soundness:** 2
**Presentation:** 1
**Contribution:** 3
**Rating:** 4
**Confidence:** 2

**Summary:**

The authors examine the relationship between attention heads in transformers and human episodic memory. They demonstrate that induction heads are behaviorally, functionally, and mechanistically similar to the contextual maintenance and retrieval model (CMR) of human episodic memory. In particular, they find that CMR-like heads often emerge in the intermediate model layers and that their behavior qualitatively mirrors the memory biases seen in humans.

**Strengths:**

From a conceptual perspective, I found this a very interesting idea, trying to link recent results from mechanistic interpretability and older results from cognitive science/modeling. In general, I think there is solid scope for such insights and information transfer. Due to that, the general framing was also well-motivated.

**Weaknesses:**

While I think there is something interesting at the basis of this work, I don’t believe that it is ready yet in its current shape. The writing, in particular, was very hard to follow, thus making it hard for me to judge the actual contents of the paper. A particularly important question to ask in this context is: who is the target audience? There are only very few people (if any) with the right background knowledge to follow the article in its present form. You pretty much find no one well-versed in mechanistic interpretability, cognitive modeling, and neuroscience. I would consider myself to have a strong background in cognitive modeling and reasonably solid knowledge of mechanistic interpretability but I was lost.

To give two concrete examples of how the writing could be improved:

(1) K-composition is introduced in Section 3.3. However, it is essentially completely irrelevant to the paper, and hence just confusing the reader.

(2) Attention scores are a crucial concept but they are not defined anywhere.

**Questions:**

The paper mentions CMR-fitted scores. What are these exactly?

Is the y-axis label in Figure 7C correct? Shouldn’t it be matching scores instead of CMR distance?

Looking beyond an exchange of ideas between two fields, are there any direct practical consequences of the discovered connection?

**Limitations:**

They are discussed appropriately.

---

> ### Author Rebuttal · Authors · 2024-08-07
>
> Thank you for acknowledging the potential of our work and offering us the opportunity to clarify. Below is our point-by-point response.
>
> ### Weaknesses
> Thank you for raising the point about the target audience of this paper. Please see the global rebuttal, point 1. Given the interdisciplinary nature of this work,  our aim is to offer new insights to readers knowledgeable in mechanistic interpretability, cognitive modeling, and/or neuroscience. Thus, we want to ensure that readers from these various fields (and hopefully beyond them!) will find our paper clear and accessible. To answer your specific points:
>
> First, we intentionally included K-composition for readers from the mechanistic interpretability (MI) community. K-composition is widely studied (Elhage et al., 2021; Singh et al., 2024; Edelman et al., 2024, Ferrando et al. 2024), while Q-composition is rarely mentioned (see Appendix of Elhage et al. 2021). Since we compare CMR with Q-composition, these readers, who are mainly familiar with K-composition, may be particularly interested in the differences between K-composition and Q-composition/CMR. We agree with you that it can be confusing for readers who are less familiar with MI. We will include this motivation in Section 3.3 and inform the reader in the Introduction which section(s) may be the most relevant based on their background and interest.
>
> Second, the attention scores we used are standard in the field, and commonly used in both the general ML literature (e.g., Dai et al., 2019) and MI (e.g., Elhage et al., 2021). However, we recognize that an explicit definition may be helpful for those from the psychology/neuroscience side: specifically, an attention score with respect to a specific token refers to the dot product of its query vector and the key vector of a (possibly different) token, scaled by the dimension of the key/query vector. In line 102, we mentioned that “We recorded the attention scores of each head (before softmax)...”, referring to the term inside the softmax function of attention as defined in Vaswani et al. (2017).
>
> ### Questions
> 1. We mentioned CMR distance and CMR-fitted attention scores (q) in line 219 (Section 5.1) and referred to Appendix C.3. We visualized a subset of them in Fig. 5a-d. Essentially, each set of CMR-fitted scores corresponds to a CRP curve produced by setting the parameters of CMR to specific values, minimizing its CMR distance to the corresponding average attention scores.
> 2. The y-axis label is correct. To clarify, top induction heads should have high matching scores and low CMR distances. We plotted CMR distance to show the difference between CMR-like heads and induction heads, which we showed an example of in Fig. 5d and discussed in the second paragraph of Discussion. We will add this clarification in the main text to avoid confusion.
> 3. Thank you for this important question. We believe that the discovered connections have many important consequences for both fields. In a broad sense, both fields have benefited tremendously in the past from the discovery of similar connections, including the relevance of Hopfield networks, convolutional neural networks, recurrent neural networks, reinforcement learning models in neuroscience and machine learning. So far, a similar connection has been largely lacking for transformers, which many researchers in neuroscience, cognitive science, and deep learning view as being fundamentally different from the brain. In that context, the evidence we provide paves the way for similar translations between the current generation of AI algorithms and a century of research in human memory.
>
>     Besides this broad connection, we can also provide specific examples of how our work might inform future research in both fields. From the perspective of mechanistic interpretability, one main contribution of our results is the mechanistic and behavioral characterization of a group of heads that may support emergent ICL abilities. One interesting implication of it is that the “lost in the middle” phenomenon seen in LLMs may be related to these heads, as humans also exhibit the same recall pattern. Understanding the connection could therefore inform ways to mitigate the problem by adopting known cognitive strategies – for example, adjusting the study schedule based on the serial position (Murphy et al., 2022). From the perspective of neuroscience, our results also suggest a common principle that underlie natural and artificial intelligence. As episodic mechanisms captured by CMR have been posited to support adaptive control and general decision-making (Lu et al., 2024; Giallanza et al., 2024; Zhou et al., 2023), this connection directly enables researchers to develop alternative mechanisms for episodic memory and its interactions with other cognitive functions based on more complex attention-head composition mechanisms (e.g., see the method analyzing N-th order virtual attention head in Elhage et al., 2021). The Discussion will spell out these two implications in more detail.
>
> **References**
> - Elhage et al. (2021). A mathematical framework for transformer circuits.
> - Singh et al. (2024). What needs to go right for an induction head?
> - Edelman et al. (2024). The evolution of statistical induction heads: In-context learning markov chains.
> - Ferrando et al. (2024). A primer on the inner workings of transformer-based language models.
> - Dai et al. (2019). Transformer-XL: Attentive Language Models beyond a Fixed-Length Context.
> - Vaswani et al. (2017). Attention is All You Need.
> - Murphy et al. (2021). Metacognitive control, serial position effects, and effective transfer to self-paced study.
> - Lu et al. (2024). Episodic memory supports the acquisition of structured task representations.
> - Giallanza et al. (2024) Toward the Emergence of Intelligent Control: Episodic Generalization and Optimization.
> - Zhou et al. (2023). Episodic retrieval for model-based evaluation in sequential decision tasks.

---

> > ### Comment · Reviewer_uMwU · 2024-08-11
> >
> > Thanks a lot for your reponse. I have increased my score to 4 but also acknowledge my high uncertainty. To make a more certain judgement, I would need to see the fully revised paper (which sadly is not possible in this review process).

---

> > > ### Author Response · Authors · 2024-08-13
> > >
> > > We appreciate your acknowledgment of our response and the increase in your score. We understand the limitations of the review process. Our final revision will address all the concerns raised, integrating the clarifications provided in our rebuttal. We are committed to providing clear value to the community.

---

### Official Review · Reviewer_xjvL · 2024-07-11

**Soundness:** 3
**Presentation:** 4
**Contribution:** 3
**Rating:** 7
**Confidence:** 4

**Summary:**

This paper explores connections between in-context learning (ICL) capabilities of large language models (LLMs) and human episodic memory. Specifically, the authors draw parallels between induction heads in Transformers and the Contextual Maintenance and Retrieval (CMR) model of human episodic memory. They demonstrate behavioural and mechanistic similarities and show that CMR-like attention patterns emerge in intermediate layers of LLMs during training. The work provides a novel and a very interesting (in my opinion at least) perspective on understanding ICL mechanisms through the lens of cognitive science models.

I recommend weak acceptance, contingent on addressing some of the weaknesses identified, particularly regarding generalizability.

**Strengths:**

1. Novelty: The paper presents an original connection between two previously separate areas of research - mechanistic interpretability of LLMs and cognitive models of human memory. I think that this approach offers insights in both fields.

2. Thorough analysis: The authors provide a detailed comparison between induction heads and CMR, including both mechanistic and behavioural analysis. The step-by-step mapping between CMR and Q-composition induction heads is particularly illuminating.

3. Empirical support: The paper includes interesting experiments on a few pre-trained LLMs (GPT-2 and Pythia models of various sizes) to support their claims. The analysis of how CMR-like behaviours emerge during training adds valuable insights.

4. Clear exposition: The paper is well-written and logically structured, making it accessible to readers from both machine learning and cognitive science backgrounds.

5. Broader impact: As in my first point, I think the work opens up new avenues for research in both AI interpretability and cognitive science, potentially leading to improved understanding in both fields.

**Weaknesses:**

1. Limited scope of experiments: The experiments focus on a specific prompt design (repeating random tokens) which may not fully capture the complexities of natural language processing. While this is acknowledged as a limitation, it would strengthen the paper to include some analysis on more naturalistic language tasks.

2. Lack of causal analysis: While the paper shows correlations between CMR-like behaviours and model performance, it doesn't establish a causal link. It remains unclear whether these behaviours are necessary for ICL or merely a byproduct of training.

3. Generalizability: The findings are primarily based on GPT-2 and Pythia models. It's not clear how well these results generalize to other Transformer architectures or non-autoregressive models. Testing on a broader range of models would strengthen the claims. Why for example use computational resources for a Pythia-12b model and not for much more advanced and well-known models, such as Llama3-8B, Mistral-7B or Qwen-7B? All of those models are also available in TransformerLens, and I believe this would make this work more relevant to the community.

4. Limited exploration of biological plausibility: While the paper draws interesting parallels to human memory, it doesn't deeply explore the biological plausibility of the proposed mechanisms. A more thorough discussion of how these findings relate to neural implementations of episodic memory would enhance the paper's impact.

5. Lack of quantitative comparison to human data: The paper shows qualitative similarities to human memory biases, but doesn't provide quantitative comparisons to human behavioural data. Such comparisons could further validate the proposed connections, although I understand that this might be a bit too much for a single NeurIPS submission.

**Questions:**

I have a few questions/suggestions for the authors:

1. Have you considered the connection between your results and neural-network models of episodic memory? A discussion on this (as well as the next comments) would be great.

2. How might the findings change if applied to more complex, natural language tasks rather than repeated random tokens?

3. Can you provide any insights into whether CMR-like behaviours are causally necessary for ICL, or if they might be an epiphenomenon?

4. How do you think these findings might generalize to non-autoregressive models or other Transformer variants?

5. Could you elaborate on potential neural implementations of the proposed mechanisms and their biological plausibility?

6. Have you considered comparing your model's behaviour quantitatively to human behavioural data on episodic memory tasks?

Also, I have a couple of minor comments:
- When describing the residual stream, I wouldd add one of the Schmidhuber citations mentioned in the Antropic paper cited here that originally described this idea (e.g. Highway Networks, Srivastava et al., 2015).
- Leave a space before the citation in line 73.
- Line 121 Tab. tab:comparison. I think you forgot to use /ref{.}.
- Letter labels b, c and d in the caption of Figure 1 as well as all letters in Figure 2 and 3 are not bold. All other letters are so please be consistent.

**Limitations:**

Overall, the authors' transparency about limitations is commendable and aligns well with the NeurIPS guidelines, although there is still room for improvement. The authors have partially addressed limitations in their Discussion section, acknowledging some key issues such as the use of random tokens as input instead of natural language, limited mechanistic interpretability offered by CMR for larger models, and potential lack of generalization to untested Transformer models. However, several important limitations identified in the paper's weaknesses are not fully addressed (see weaknesses). Potential negative societal impacts are deemed minimal for this type of foundational research.

---

> ### Author Rebuttal · Authors · 2024-08-07
>
> Thank you for your interest and thoughtful feedback on our work. Below is our point-by-point response:
> ### Weaknesses
> 1. We used repeated random tokens because: (1) it aligns with human free recall experiments, where random words are presented sequentially (Murdock, 1962); (2) it is a widely acknowledged definition of induction heads in mechanistic interpretability literature (Elhage et al., 2021; Olsson et al., 2022; Bansal et al., 2022; Nanda, 2022; Crosbie et al., 2024); (3) it uses off-distribution prompts to focus on abstract properties, avoiding potential confounds from normal token statistics (Elhage et al., 2021). However, understanding these CMR-like heads in naturalistic language tasks is important, so we include a new analysis in the global rebuttal (point 4).
> 2. Thank you for the suggestion. Please see the causal analysis in the global rebuttal (point 4).
> 3. Thank you for the suggestion on generalizability. Please see the global rebuttal (point 3), where we replicated the results on the three models suggested. We chose the Pythia series due to their shared architecture and training checkpoints, which are informative about the timeline when ICL abilities and CMR-like heads emerge.
>
>     We focused on autoregressive models with causal attention because their induction heads are widely studied, and because CMR is autoregressive and causal. Whether the biological brain has a non-autoregressive objective (e.g., masked language modeling) is an open question. In-context learning is less studied in non-autoregressive models like BERTs (but see Samuel 2024), and little is known about their “induction heads.” How the behavior and mechanisms of induction heads in GPT-like models generalize to BERT-like models is an open research question.
>
> 4. While CMR is a behavioral model, neuroscience suggests it can also explain patterns of neural activity. In CMR, episodic retrieval occurs in two phases: (1) retrieving a word based on the current temporal context via matrix $\mathbf{M}^{\rm TF}$; (2) retrieving the temporal context associated with the word via matrix $\mathbf{M}^{\rm FT}$. These associative matrices represent the full set of episodic memories, instantiated in hippocampal synapses. The temporal context serves as both input and output of the retrieval process in CMR, aligning with the hippocampus's recurrent nature. Studies suggest the temporal context is represented in hippocampal subregions (e.g., CA1 and dentate gyrus, Sakon & Kahana, 2022; Dimsdale-Zucker et al., 2020, with $\mathbf{M}^{\rm TF}$ and $\mathbf{M}^{\rm FT}$ in CA3). Others propose the temporal context is represented in cortical regions providing input to the hippocampus (e.g., entorhinal cortex; Howard et al., 2005), with $\mathbf{M}^{\rm TF}$ and $\mathbf{M}^{\rm FT}$ in the broader hippocampal network. We will include this in the revised manuscript.
> 5. We agree that human data should be included to contextualize our findings. To this end, we added typical CRP curves from human studies (Fig. R1a) and a more extensive comparison of fitted parameters between top CMR-like heads and average human subjects from previous experiments (Fig. R1b). Please see the global rebuttal (point 2) and Fig. R1.
> ### Questions
> 1. We see rich connections between our results and neural network models of episodic memory, which can benefit research in both directions. For example, Salvatore & Zhang (2024) found that when trained to maximize recall accuracy, a seq2seq model with attention shows the same recall pattern as the best-performing CMR, with intermediate states showing similar recall patterns to human subjects. Li et al. (2024) found that RNNs trained for free recall produced the same recall order as the optimal CMR and the method of loci (where people mentally “place” items in imagined locations and then “retrieve” them in the same order), even without explicitly optimizing the recall order. Giallanza et al. (2024) showed that a neural network implementation of CMR can explain flexible cognitive control in humans. These findings suggest that CMR-like behavior can emerge in neural networks with recurrence and/or attention mechanisms, making it efficient for prediction and advantageous for general decision-making.
>
>     Our results are also consistent with work exploring connections between attention mechanisms in transformers and the hippocampal formation (Whittington et al., 2021). While that work focused on emergent place and grid cells in transformers with recurrent position encodings, the hippocampal subfields involved are also postulated to represent CMR components (see our reply to weakness, point 4). These results generally support our proposal linking the query-key-value retrieval mechanism to human episodic retrieval.
> 2. Please see our reply to weakness points 1 and 2.
> 3. Please see our reply to weakness point 2.
> 4. Please see our reply to weakness point 3.
> 5. Please see our reply to weakness point  4.
> 6. Please see our reply to weakness point 5.
>
> Finally, thank you for the minor comments. We will modify the texts/labels accordingly.
>
> **References**
> - Murdock et al. (1962). The serial position effect of free recall.
> - Elhage et al. (2021). A mathematical framework for transformer circuits.
> - Olsson et al. (2022) In-context Learning and Induction Heads.
> - Bansal et al. (2022). Rethinking the role of scale for in-context learning: An interpretability-based case study at 66 billion scale.
> - Nanda. (2022). Induction mosaic.
> - Crosbie et al. (2024). Induction Heads as an Essential Mechanism for Pattern Matching in In-context Learning.
> - Samuel. (2024). BERTs are Generative In-Context Learners.
> - Salvatore et al. (2024). Parallels between Neural Machine Translation and Human Memory Search: A Cognitive Modeling Approach.
> - Li et al. (2024). A neural network model trained on free recall learns the method of loci.
> - Giallanza et al. (2024) Toward the Emergence of Intelligent Control: Episodic Generalization and Optimization.

---

> > ### Comment · Reviewer_xjvL · 2024-08-13
> >
> > Thank you for your comprehensive and thoughtful rebuttal. I appreciate the additional analyses you've conducted, particularly on more recent models like Llama3-8B, Mistral-7B, and Qwen-7B, as well as the ablation study demonstrating the causal link between CMR-like heads and in-context learning performance. Your inclusion of human behavioral data comparisons and expanded discussion on biological plausibility have also strengthened the paper significantly.
> >
> > I believe the other reviewers have raised valid concerns, particularly regarding the clarity of Sections 3 and 4. I strongly advise revising these sections based on the collective feedback we've provided. Given your thorough responses and plans for improvement, I trust you will address these issues effectively.
> >
> > Considering these planned improvements, I am increasing my score to 7 (Accept). In case the paper gets accepted, I believe all these changes (if implemented in the final version) will make your work more impactful and valuable to the NeurIPS community. Good luck!

---

> > > ### Author Response · Authors · 2024-08-13
> > >
> > > Thank you for your thorough review and constructive feedback throughout this process. We're grateful for your recognition of our additional analyses and the strengthening of our paper. We appreciate your guidance and we commit to implementing these revisions effectively.

---

### Official Review · Reviewer_rPH1 · 2024-07-12

**Soundness:** 2
**Presentation:** 2
**Contribution:** 3
**Rating:** 4
**Confidence:** 2

**Summary:**

The paper compares LLMs to a neuroscience model of human episodic memory, particularly highlighting the similarities between induction heads (responsible for in-context learning in transformers) and CMR.

**Strengths:**

1. The work is original and offers a deeper understanding of induction heads by linking them to a well-studied model of human memory.
2. The potential significance is high, but improvements in analysis and clarity are needed for the community to build on these ideas.

**Weaknesses:**

1. The paper lacks clarity, making it difficult to understand.
2. The analysis has several issues, which will be discussed in the limitations section.

**Questions:**

I have mentioned them in the limitations section below.

**Limitations:**

While the idea of the paper is promising, there are several points in the analysis that need to be addressed. I am open to changing my score if these points are addressed with new analysis.
1. **Clarity**: Section 4.1, explaining the CMR model, needs to be rewritten for better understanding, especially since the ML community might not be familiar with this model.
2. **Comparison to Human Data**: The paper claims similarities to human memory but doesn't compare its results with human data. Including such comparisons would illustrate how the temporal contiguity and forward asymmetry biases appear in humans.
3. **Hyperparameter fitting**: Fitting that many hyperparameters to each layer could give good fits to so many models and therefore does not prove anything. It lacks hypothesis testing or baselines. For instance, you argue that it is similar to Q-composition-like heads, and therefore fitting the model to Q-compositions vs K-compositions and showing the difference in fits would give more value to this CMR distance and interpretation. Because I can just choose any model from any neuroscience litterature and just fit it and I believe it will give me some good fits with enough degree of freedom. Therefore, some baselines would be interesting. The reader needs a reference.
4. **Hyperparameter interpretation**: The meaning of hyperparameters needs more exploration. The only mention of this is the discussion of temporal clustering with the β values which is quite vague. For instance, you say that in Figure 7 "The increase in β_rec  was particularly prominent, indicating the importance of temporal clustering during decoding for model performance." which from my interpration seem to say that it is particularly prominent relative to β_end, which does not seem visually to be the case. But maybe I missed something.
5. **Qualitative Trends** As for the above point, I felt like sometimes you just cherry pick some qualitative trends that are supposed to match your claims and they are not even clear. One example is line 227: "we found that the majority of heads in the intermediate layers of GPT2-small have lower CMR distances (Fig. 6a). " Again, maybe I got this wrong but this is really not what the figure shows, it shows a linear increase across layers.
6. The CMR distance < 0.5 grouping seems arbitrary. Do you get similar results with < 0.1 and < 1 for instance?
7. Could you add error bars in Figure 7?

---

> ### Author Rebuttal · Authors · 2024-08-07
>
> Thank you for recognizing the value of our work and offering constructive feedback. We provide a point-by-point response below.
>
> ### Weaknesses
> Please see below.
>
> ### Limitations
> 1. We recognize the need for more clarity, particularly for those unfamiliar with CMR. In the revised manuscript, we will enhance Section 4.1 with a more intuitive explanation and add a pedagogical description in the appendix. Please refer to our global rebuttal (point 1) for details.
>
> 2. We agree that including human data is crucial for context. The revision will feature typical CRP curves from human studies (Fig. R1a) and a more comprehensive comparison of fitted parameters between top CMR-like heads and average human subjects (Fig. R1b). See our global rebuttal (point 2) and Fig. R1 for more information.
>
> 3. Thank you for the suggestion. As a baseline, we have included a descriptive model using Gaussian functions (with the same number of parameters as CMR). Its bell shape captures the basic aspects of temporal contiguity and forward/backward asymmetry. We found that, across 12 different models (GPT2, all Pythia models, Qwen-7B, Mistral-7B, Llama3-8B), CMR provides significantly better descriptions (lower distances) than the Gaussian function for the top induction heads (average CMR distance: 0.11 (top20) / 0.05 (top50) / 0.12 (top100) / 0.12 (top200), average Gaussian distance: 1.0 (top20) / 0.98 (top50) / 0.98 (top100) / 0.97 (top200), all p<0.0001). Importantly, Gaussian functions offer little, if any, insight into how these attention patterns might arise in the first place. In contrast, our CMR shows a direct link to the Q-composition induction head at the mechanistic level.
>
>     Additionally, we are not aware of other substantially different models from the broad neuroscience literature that can give good – let alone meaningful – results as ours. Model-free reinforcement learning and other models without temporal components will perform poorly, regardless of the degree of freedom. Even with a temporal component, models like drift-diffusion models or random context models (Murdock, 1997) cannot capture the attention patterns of induction heads, as their CRP will be essentially flat (Howard & Kahana, 2002). In fact, CMR is well-poised to explain the properties of observed CRPs, because previous models failed to explain the asymmetric contiguity, and all current successful models share the same core dynamics as CMR. If you have a particular alternative model in mind that might provide insights into the mechanisms behind induction heads and better explanations than CMR, we are happy to perform a more specific comparison.
>
>     Finally, to improve our results over correlational model-fitting, we performed an ablation study and found that removing heads with the lowest CMR distances causes significantly worse ICL performance, suggesting they are in fact necessary for ICL (see global rebuttal point 4 and Fig. R5).
>
> 4. Thank you for pointing out the unclear sentence. We have revised it to: “The training process leads to higher values of $\beta_\text{rec}$. Specifically, $\beta_\text{rec}$ values are higher than $\beta_\text{enc}$, highlighting the importance of temporal clustering during decoding for model performance.”
>
> 5. We appreciate the concerns and believe the revisions demonstrate that our results are not cherry-picked. In particular, we have changed the sentence in line 227 to:“... we found that the majority of heads in the intermediate and later layers of GPT2-small have lower CMR distances than earlier layers”. This is supported by the distribution of CMR distances of heads in each layer (Fig. R2a). Additionally, it is inaccurate to claim “a linear increase across layers” given the low proportion (Fig. 6a) and high CMR distance (Fig. R2a) for early-intermediate layers (~25% position).  Furthermore, heads with lower CMR distances tend to occur in intermediate-late layers (rather than earlier or later layers) across twelve models of varying complexity (Fig. R2b). We discuss more evidence in the next response.
>
> 6. With a goal to identify heads with CMR-like attention scores, the threshold of 0.5 is informed by both the quantitative distribution of individual head CMR distances and exploratory qualitative analysis of the attention patterns. First, as the bottom panel of Fig. 5e shows, the distribution of individual head CMR distances in GPT2 can be divided roughly into two clusters: a cluster peaking around 0.1 (and extending to around 1), and a spread-out cluster around 2.4. We examined heads with ~1 CMR distance and found non-human-like CRP (e.g., Fig. R3a, b). Further, many heads with CMR distance around 0.6-0.8 again show CRP different from humans (e.g., Fig. R3c, d). Therefore, we operationally set the threshold of 0.5. We also tested a threshold of 0.1 and found no qualitative difference (Fig. R2c, d). Thus our threshold is chosen to balance the inclusion of CMR-like heads and the exclusion of non-CMR-like heads. We will include this clarification in the appendix.
>
> 7. Please refer to (Fig. R4) for an updated Fig. 7 with standard errors shown.
>
> **References**
> - Elhage et al. (2021). A mathematical framework for transformer circuits.
> - Olsson, et al. (2022). In-context Learning and Induction Heads.
> - Crosbie, J., & Shutova, E. (2024). Induction Heads as an Essential Mechanism for Pattern Matching in In-context Learning.
> - Murdock, B. B. (1997). Context and mediators in a theory of distributed associative memory (TODAM2).
> - Howard, M. W., & Kahana, M. J. (2002). A distributed representation of temporal context.

---

> > ### Comment · Reviewer_rPH1 · 2024-08-12
> >
> > Thank you for the revisions. While I appreciate the efforts made, it's challenging to assess the changes in Sections 3 and 4 within the constraints of the NeurIPS rebuttal which I believe is key for acceptance. I do believe this project has a lot of scope but there is a lot of moving parts and it needs to be better merged together for the community to benefit from these insights. However, I believe the paper is now in better shape, and I am raising my score to a 4.

---

> > > ### Author Response · Authors · 2024-08-14
> > >
> > > Thank you for your feedback and for raising the score. We appreciate your recognition of the project's potential. We understand the limitations of the rebuttal process and ensure that the revised paper will address concerns about Sections 3 and 4, integrating the various components more cohesively. We are committed to ensuring our work provides clear, valuable insights to the community.

---

### Author Rebuttal · Authors · 2024-08-07

We thank the reviewers for their constructive comments. Here we respond to questions asked by multiple reviewers:
## 1 Clarity and accessibility of our paper
While some reviewers praised the clarity and accessibility of our paper, others felt that there was room for improvement. Given the interdisciplinary nature of our work, we hope to reach a broad audience in ML/mechanistic interpretability, cognitive modeling, and neuroscience. Accordingly, we will revise Sections 3 and 4 to provide more intuition about induction heads and CMR before diving into the technical details. Given the space limit, we also plan to include a more detailed, pedagogical description of induction heads and CMR in the appendix for the revised paper.

## 2 Comparison to human data
We thank the reviewers for pointing out the missing human data. We analyzed the CRPs (Fig. R1a) of the human free recall data from Zhang et al. (2023). The curves exhibit a forward asymmetry and temporal contiguity as described in Section 4.1. The top-performing subjects have a sharper CRP with a larger forward asymmetry, compared to other subjects. Correspondingly, the fitted $\beta_\text{rec}$ in CMR is higher for top-performing subjects (typically around 0.7 or higher). Similarly, our results suggest that top induction/CMR-like heads often exhibit larger $\beta_\text{rec}$.

We also performed a more extensive literature review to determine the distribution of $\beta$ observed in existing human studies (Fig. R1b). The average fitted values of $\beta_\text{enc}$ and $\beta_\text{rec}$ for human subjects vary from study to study (0.2-0.9). Our results suggest that the top CMR-like heads exhibit a degree of temporal clustering consistent with that in humans.

## 3 Generalizability of our results
We agree that verifying the generalizability of our results on more LLM models with different architectures is crucial. We performed the same analysis on three additional models suggested by Reviewer xjvL (Llama3-8B, Mistral-7B, Qwen-7B) and the findings were similar (Fig. R6). Specifically, CMR-like heads appear most frequently in intermediate-late layers (Fig. R6a), and the top heads generally have high temporal clustering (high $\beta_\text{enc}$, even higher $\beta_\text{rec}$; Fig. R6b, d).

## 4 Causal analysis of CMR-like heads on a more naturalistic language task
We appreciate the reviewers’ questions about why CMR and CMR distance should be of interest to our understanding of large language models. Our results as shown in the paper mainly suggest a correlational relationship between small CMR distances (“CMR-like heads”) and in-context learning (ICL) on repeated random tokens.

To test whether CMR-like heads are causally necessary for ICL in more naturalistic sentences, we conducted an ablation study. We ablated the top 10% CMR-like heads (i.e., top 10% heads with the smallest CMR distances) in each model and computed the resultant ICL score (evaluated on the first 1000 texts of the processed version of Google’s C4 dataset (allenai/c4 on huggingface) with at least 512 tokens, except pythia-12b, which was evaluated using the first 500 due to time constraint). Specifically, the ICL score is defined as “the loss of the 500th token in the context minus the loss of the 50th token in the context, averaged over dataset examples” (Olsson et al., 2022). Intuitively, a model with better in-context learning ability has a lower ICL score, as the 500th token is further into the context established from the beginning. We tested models with various model architectures and complexity, including GPT2, Pythia models, and Qwen-7B (Fig. R5, ranked by original ICL score from lowest to highest).

Most models (except Pythia-1B) showed a higher ICL score (worse ICL ability) if the top 10% CMR-like heads were ablated, compared to the case where the same number of randomly selected heads were ablated. This effect was particularly significant if the original model had a low ICL score (e.g., GPT2, Qwen-7B). We note, however, there are a few complications when explaining the results: we lack time to average over enough samples to further reduce the error bars; generally there might be a Hydra effect where ablation of heads causes other heads to compensate (McGrath et al 2023); the ICL scores for original models are closer to 0 (weaker ICL performance) in Pythia series than other architectures, suggesting either Pythia series’ ICL abilities are weaker, or the distributional differences between their training data and evaluation data are larger.

In short, our finding suggests that CMR-like behavior is not merely an epiphenomenon; it is essential underlying LLM’s ICL ability. CMR distance thus provides a meaningful metric to characterize individual heads in LLMs.

---

### Author Response · Authors · 2024-08-13

Thank you all for your valuable feedback. We appreciate the recognition of our work's potential impact and interdisciplinary approach. In addition to the analyses in the rebuttal, we acknowledge the concerns about Sections 3 and 4 and commit to thorough implementations of feedback to enhance accessibility. Our goal is to present a final version that addresses all concerns, improves clarity, and strengthens the paper's impact across the diverse NeurIPS community.

---

### Decision · Program_Chairs · 2024-09-25

**Decision:**

Accept (poster)

**Comment:**

This submission presents mechanistic and behavioral correspondences between transformer language models and a classical model of human episodic memory, the contextual maintenance and retrieval (CMR) model. Connections of this kind are of great potential interest to the NeurIPS community because of a growing gap between neuro-inspired and engineering-based approaches to designing machine learning systems, and this submission bridges these two subdomains nicely. The submission and author responses themselves are thorough and precise, and the authors provided open-source code. Reviewers had no qualms about the technical details of the paper but raised concerns about the readability of the paper and the causal status of the investigations. The authors responded to the latter by providing preliminary causal evidence in the author response, and promised to substantially improve the paper's readability to address the former.

Since no major extra content beyond the submission and the rebuttal will be required to address the reviewers' main concerns, I am inclined to accept this paper but request that authors indeed follow through with the promised changes, listed below for accountability (though perhaps I have missed some):

1. Improve clarity and accessibility of Sections 3 and 4.

2. Include comparisons to human behavioral data and evaluations with more models presented in the global response.

3. Address the causal analysis more carefully.

4. Expand the discussion on practical implications of their findings for both mechanistic interpretability and neuroscience fields.

 I also request the following improvements in formatting for the camera-ready:

- Figures at the top of the page, rather than breaking up text.

- Better global placement of figures near their relevant text, especially Figure 3.

- Some other formatting solution for Section 4.2 than run-in numbering, which makes this poorly readable as a recipe.